# SYMBOL: GENERATING FLEXIBLE BLACK-BOX OPTIMIZERS THROUGH SYMBOLIC EQUATION LEARNING

**Jiacheng Chen**[1,*]**, Zeyuan Ma**[1,*]**, Hongshu Guo**[1]**, Yining Ma**[2]**, Jie Zhang**[2]**, Yue-Jiao Gong**[1,†]

[1] South China University of Technology        [2] Nanyang Technological University

{jackchan9345, scut.crazynicolas, guohongshu369}@gmail.com,
{yining.ma, zhangj}@ntu.edu.sg ,gongyuejiao@gmail.com

## ABSTRACT

Recent Meta-learning for Black-Box Optimization (MetaBBO) methods harness neural networks to meta-learn configurations of traditional black-box optimizers. Despite their success, they are inevitably restricted by the limitations of predefined hand-crafted optimizers. In this paper, we present SYMBOL, a novel framework that promotes the automated discovery of black-box optimizers through symbolic equation learning. Specifically, we propose a Symbolic Equation Generator (SEG) that allows closed-form optimization rules to be dynamically generated for specific tasks and optimization steps. Within SYMBOL, we then develop three distinct strategies based on reinforcement learning, so as to meta-learn the SEG efficiently. Extensive experiments reveal that the optimizers generated by SYMBOL not only surpass the state-of-the-art BBO and MetaBBO baselines, but also exhibit exceptional zero-shot generalization abilities across entirely unseen tasks with different problem dimensions, population sizes, and optimization horizons. Furthermore, we conduct in-depth analyses of our SYMBOL framework and the optimization rules that it generates, underscoring its desirable flexibility and interpretability.

## 1 INTRODUCTION

Black-Box Optimization (BBO) pertains to optimizing an objective function without knowing its mathematical formulation, essential in numerous domains ranging from automated machine learning (Akiba et al., 2019) to bioinformatics (Tsaban et al., 2022). Due to the black-box nature, optimizing BBO tasks requires a search process that iteratively probes and evaluates candidate solutions. While several well-known BBO optimizers, such as Differential Evolution (Storn & Price, 1997), Evolutionary Strategy (Hansen & Ostermeier, 2001), and Bayesian Optimization (Snoek et al., 2012b) have been proposed to steer the search process based on hand-crafted rules, they often necessitate intricate manual tuning with deep expertise to secure the desirable performance for a particular class of BBO tasks.

To mitigate the labour-intensive tuning required for BBO optimizers, recent Meta-Black-Box Optimization (MetaBBO) methods have turned to meta-learning (Finn et al., 2017) to automate such process. In the left portion of Figure 1, we depict existing prevailing BBO (in the first block) and MetaBBO (in the remaining blocks) paradigms. Essentially, MetaBBO follows the bi-level optimization and can be categorized into two branches. The first branch is the *MetaBBO for auto-configuration*. As shown in the second block, at the meta level, a neural policy dictates configurations for the lower-level BBO optimizer, and this lower-level optimizer typically employs an expert-derived formulation; Together, they strive to optimize a meta-objective to excel across a broader range of BBO tasks. While they offer a pathway to data-centric optimization, the fine-tuned optimizer closely follows the designs of the backbone BBO optimizer, thereby inheriting the potential limitations of human-crafted rules. More recently, some MetaBBO methods employ neural networks to directly propose subsequent candidate solutions shown in the third block, however, these models may easily get overfitted and their efficiency may be limited to low-dimensional BBOs only.

---

[*]Equal Contribution
[†]Corresponding Author

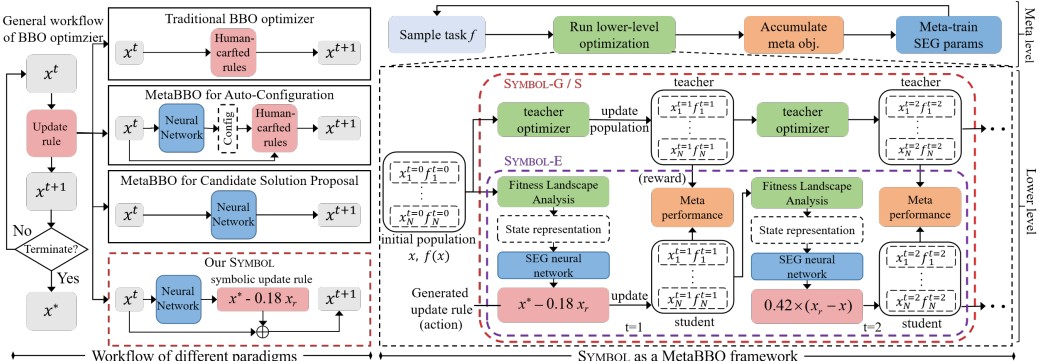

Figure 1: Comparison of traditional BBO, MetaBBO, and our SYMBOL frameworks. **Left**: Depiction of various BBO methods. **Right**: A view of the bi-level structure of SYMBOL. The lower level optimizes individual BBO tasks, with the SEG generating step-by-step update rules. The meta level compiles performances from the lower level into a meta objective so as to learn the SEG through either the SYMBOL-E (without a teacher BBO optimizer), SYMBOL-G, or SYMBOL-S strategy.

Meanwhile, while they attempt to bypass hand-crafted rules, they shift towards becoming "black box" optimization systems, raising concerns about interpretability and generalization.

In this paper, we address the above challenges by presenting **SYMBOL**, a novel MetaBBO framework designed to autonomously generate **sym**bolic equations for **b**lack-box **o**ptimizer **l**earning. As shown in the final block on the left side of Figure 1, our proposed SYMBOL considers generating update rules expressed as closed-form equations, which not only allows SYMBOL to minimize its dependence on hand-crafted optimizers, enhancing flexibility for discovering novel BBO optimizers, but also to exhibit much better interpretability. Specifically, our SYMBOL framework, illustrated in the right portion of Figure 1, follows the bi-level structure of MetaBBO. It features a *Symbolic-Equation-Generator* (SEG) at the lower level to dynamically generate symbolic update equations for optimizing a given BBO task and a given optimization step. The generated rules then determine the next generations of candidate solutions (or the next population in BBO parlance). At the meta level, the SEG is trained to maximize the meta-objective across various BBO tasks.

Nevertheless, achieving the above flexible and interpretable framework certainly comes with its challenges. To this end, we introduce the following novel designs: 1) At the lower level, our SEG autoregressively generates closed-form equations with each symbol drawn from a basis symbol set rooted in first principles; concurrently, we also design a constant inference mechanism for incorporating constants in the equations; 2) At the meta level, we put forth three unique SYMBOL training strategies to meta-learn SEG, namely the Exploration strategy (SYMBOL-E), which initiates the learning from scratch; the Guided strategy (SYMBOL-G), which harnesses a state-of-the-art BBO optimizer as a teacher for behavior cloning; and the Synergized strategy (SYMBOL-S), which amalgamates the previous two for effective training; 3) Lastly, to bolster zero-shot generalization, our SEG further combines fitness landscape analysis with a robust contextual representation, while our SYMBOL leverages a diverse training set that encompasses various landscape properties.

Through benchmark experiments, we validate the effectiveness of our SYMBOL framework, demonstrating its efficacy in discovering optimizers that surpass the existing hand-crafted ones. Moreover, SYMBOL stands out with its zero-shot generalization ability, matching the performance of well-known SMAC optimizer (Lindauer et al., 2022) but with faster speed on the entirely unseen BBO tasks such as hyper-parameter optimization and protein docking. We also discuss the optimization rules that our SYMBOL generates. Lastly, we conduct extensive analyses of SYMBOL regarding its teacher options, hyper-parameter sensitivities, and ablation studies. Our main contributions are four folds: 1) The introduction of SYMBOL, a pioneering generative MetaBBO framework for automated optimizer discovery; 2) The SEG network, designed to effectively generate closed-form BBO equations; 3) Three training strategies based on reinforcement learning for different application needs; and 4) New state-of-the-art MetaBBO performance complemented by in-depth discussions.

## 2 RELATED WORKS

**Traditional black-box optimizer.** Traditional black-box optimizers seek global optima without relying on gradient information, with population-based algorithms being particularly dominant. Algorithms like Genetic Algorithm (GA) (Holland, 1992), Differential Evolution (DE) (Storn & Price, 1997), Particle Swarm Optimization (PSO) (Kennedy & Eberhart, 1995) focus on the inner-population evolution, ensuring convergence of individual solutions within the population towards the global optima. Others, like Evolution Strategy (ES) (Hansen et al., 2003), focus on refining the population distribution based on statistics. To further enhance performance, adaptive strategies have been developed, including dynamic population size strategy (Biswas et al., 2021; Auger & Hansen, 2005), multi-operator selection strategy (Brest et al., 2021), adaptive control parameter strategy (Sarker et al., 2013), etc. Besides population-based methods, Bayesian Optimization (BO) (Snoek et al., 2012a; 2015) leverages Gaussian Processes and posterior acquisition to discern optimal solutions. Among these, SMAC (Lindauer et al., 2022) makes advances with a racing mechanism, setting the state-of-the-art performance in autoML hyper-parameter optimization (Arango et al., 2021).

**Meta-learned black-box optimizer.** Initial MetaBBO research utilizes traditional black-box optimizers at the meta level to determine proper configurations for lower-level optimizers, where a single configuration is usually generated for the entire optimization process (Zhao et al., 2023). Recent MetaBBO meta-learns the configurations or update rules of black-box optimizers using a neural network, aiming to generate more flexible update rules dynamically. Typically, they consider fine-tuning the configurations of a backbone BBO optimizer, e.g., auto-selecting evolution operators and hyper-parameters for the evolutionary optimizers (Sharma et al., 2019; Sun et al., 2021; Lange et al., 2023; Chaybouti et al., 2022; Lange et al., 2022; G.Shala et al., 2020; Li et al., 2023; Wu et al., 2023). Though boosting the performance, they remain constrained by the inherent limitations of the backbone optimizer. Another way is to directly suggest the next candidate solution(s) through neural networks (Chen et al., 2017; TV et al., 2019). These end-to-end models can match the performance of state-of-the-art black-box optimizers in low-dimensional tasks ($\leq 6$). Yet, for larger scales, their efficiency can diminish. Furthermore, their shift towards becoming "black box" systems raises interpretability concerns. Meanwhile, we note that many of the above MetaBBO works may still fail to generalize across different task distributions (Sharma et al., 2019), dimensions (Chen et al., 2017; TV et al., 2019; Chaybouti et al., 2022; Cao et al., 2019), population sizes (Sun et al., 2021), or optimization horizons (Chen et al., 2017; TV et al., 2019). Symbolic-L2O (Zheng et al., 2022) and Lion (Chen et al., 2023), using symbolic learning, show a certain similarity with our SYMBOL in concept. But it focuses on discovering gradient descent optimizers, whereas SYMBOL is designed to devise novel update rules for a range of BBO tasks.

**Symbolic equation learning.** Symbolic equation learning aims to discover a mathematical expression that best fits the given data, which is a core concept in the realm of Symbolic Regression (SR). Traditional SR algorithms utilize genetic programming (GP) to evolve and optimize symbolic expressions (Schmidt & Lipson, 2009), whereas recent studies embrace the assistance of neural-guided search methodologies, such as utilizing a Recurrent Neural Network (RNN) combined with reinforcement learning, as in (Petersen et al., 2021). In this approach, RNN is used to autoregressively infer a function $\hat{f}$, and the reward could be calculated as the error between the ground truth $f(x)$ and the predicted values $\hat{f}(x)$. The constants in $\hat{f}$ are local-searched by the Broyden–Fletcher–Goldfarb–Shanno (BFGS) (R.Fletcher, 1970), performing iterative evaluations and adjustments. In our proposed SYMBOL, we favor using RNN to formulate symbolic update rules over resorting to GP-driven search, primarily due to two reasons: 1) the intensive computational demands of GPs, and 2) its pronounced sensitivity to hyper-parameter choices (Biggio et al., 2021).

## 3 METHODOLOGY

Our SYMBOL framework, akin to MetaBBO, follows bi-level optimization. Starting with an initial population (a set of solutions), the Symbolic Equation Generator (SEG) at the lower level dynamically formulates the update rules, $\tau^{(t)}$, for advancing the populations (current solutions) from $x^{(t)}$ to $x^{(t+1)}$ using the formula $x^{(t+1)} = x^{(t)} + \tau^{(t)}$. Here, we use $t$ to represent the optimization step (generation) and we depict the whole process in the bottom right of Figure 2. In the rest of Figure 2, we provide an example of how SEG autoregressively generates such update rules $\tau^{(t)}$ based on a *basis symbol set* $\mathbb{S}$ (at the top) and the on-the-fly constant value inference mechanism (highlighted

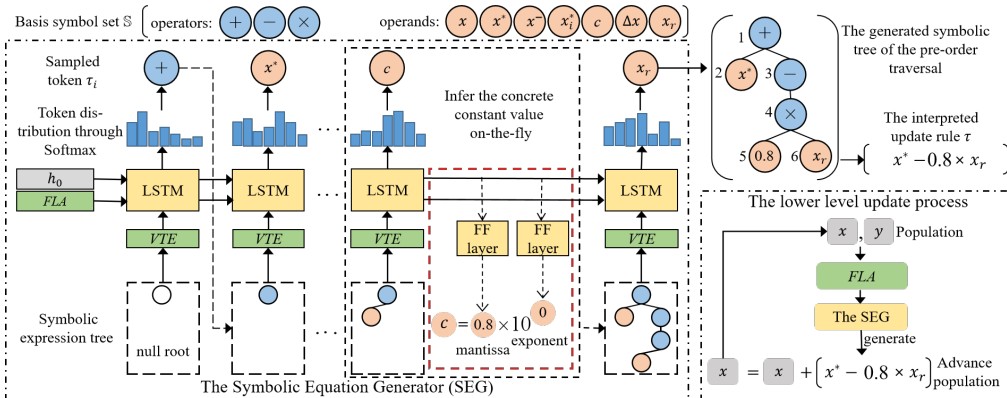

Figure 2: The illustration of our symbolic equation generator (SEG) at the lower level. **Left:** An example of generating the update rule $\tau$ via SEG with on-the-fly inference of constant $c$ (highlighted in red dashed block). **Top right:** The symbolic expression tree from the pre-order traversal and the interpreted update rule $\tau$. **Bottom right:** an illustration of the overall iterations of SEG.

in the middle). At the meta level, the SEG is trained via three different strategies of SYMBOL to maximize the meta-objective over a task distribution.

## 3.1 SYMBOLIC EQUATION GENERATOR

Our SEG leverages the structure of symbolic expression trees to represent analytical equations (Petersen et al., 2021), with internal nodes denoting operators (e.g., $+$) and terminal nodes signifying operands (e.g., constants). Starting with an empty expression tree, it involves autoregressively selecting a token from a *basis symbol set* $\mathbb{S}$, so as to construct the pre-order traversal of this symbolic expression tree. The traversal can then be uniquely interpreted into the corresponding update rule with a closed form. In this paper, we adopt $\tau$ to denote the generated update rule, its symbolic expression tree, and the corresponding pre-order traversal, interchangeably. We denote $\tau_i$ as the $i^{\text{th}}$ token of $\tau$ and the depth of the expression tree as $H$.

We parameterize the SEG with an LSTM, denoted as $\theta$, to autoregressively construct update rules token by token. In each construction step, we input the LSTM with a Vectorized Tree Embedding (VTE) of the partially constructed expression tree, complemented by features originating from Fitness Landscape Analysis (FLA) as the initial cell state of the LSTM so as to capture the current optimization status. The output would be a categorical distribution over all the available tokens from the basis symbol set $\mathbb{S}$, where each token $\tau_i$ is sampled from the distribution, during both training and inference. This process terminates either when all terminal nodes are assigned as operands or when the depth of the tree, $H$, exceeds a predetermined limit. The likelihood of the pre-order traversal, $\tau$, is the cumulative product of the probabilities of selecting each token, i.e., $p(\tau|\theta) = \prod_{i=1}^{|\tau|} p(\tau_i|\tau_{1:(i-1)}, \theta)$. More details are provided as follows.

**Basis symbol set** $\mathbb{S}$**.** Following the first principle, we identify an essential symbol set $\mathbb{S}$, i.e., $\{+, -, \times, x, x^*, x^-, x_i^*, \Delta x, x_r, c\}$, where we divide them into operators and operands. 1) Operators: $+$ $-$ and $\times$. 2) Operands: $x$ - 'candidate solutions', $x^*$ and $x^-$ - 'the best and worst solutions found so far in the whole population', $x_i^*$ - 'the best-so-far solution for the $i^{\text{th}}$ candidate', $\Delta x$ - 'the differential vector (velocity) between consecutive steps of candidates', $x_r$ - 'a randomly selected candidate from the present population of candidates', and $c$ - 'constant values'. Such basis symbol set covers update rules for known optimizers such as DE (Storn & Price, 1997) with mutation strategy $\tau = (x_{r1} - x) + c \times (x_{r2} - x_{r3})$, PSO (Kennedy & Eberhart, 1995) with update formula $\tau = c1 \times \Delta x + c2 \times (x^* - x) + c3 \times (x_i^* - x)$, etc. More importantly, $\mathbb{S}$ could also span beyond the existing hand-crafted space, offering flexibility to explore novel and effective rules. Note that the design of $\mathbb{S}$ needs to balance representability and learning effectiveness, simply removing/augmenting symbols from/to $\mathbb{S}$ may impact the final performance, which is discussed in Appendix C.2.

**Constant inference.** Upon the selection of the constant token $c$ for $\tau_i$, we infer its value immediately by processing the subsequent LSTM hidden states $h_{i+1}$ through two distinct feed-forward layers.

Specifically, any constant $c$ is defined as a combination of its mantissa $\varpi$ and exponent $\epsilon$, represented as $c = \varpi \times 10^{\epsilon}$, where we stipulated that $\epsilon \in \{0, -1\}$ and $\varpi \in \{-1.0, -0.9, -0.8, ..., 0.8, 0.9, 1.0\}$. The process is shown in Figure 2, highlighted in the red dashed block. Note that such an inference mechanism for constants grants SYMBOL the versatility to fine-tune the relative emphasis of different segments within an update rule via sampling from various exponent levels.

**Masks for streamlined equation generation.** Following past works (Biggio et al., 2021; Valipour et al., 2021; Vastl et al., 2022), we introduce several restrictions to simplify symbolic equation generation: 1) The children of binary operators must not both be constants as the result is simply another constant; 2) The generated expressions must not include consecutive inverse operation, e.g., $+x - x$; 3) The children of the operator $+$ must not be the same since it is equivalent to the operator $\times$; 4) The operator $\times$ must have one constant child because two-variable multiplication merely appears in any black-box optimizer; 5) The height $H$ of the symbolic expression tree is empirically set to be between $2$ and $5$, which aligns with most of traditional black-box optimizers. Given the above restrictions, we mask out the probability of selecting any invalid token as $0$.

**Contextual state representation.** To inform SEG of sufficient contextual information for generating update rules, our state representation includes two aspects: 1) Vectorized Tree Embedding (*VTE*), which embeds the partially constructed symbolic expression tree created thus far into a vector; and 2) Features derived from Fitness Landscape Analysis (*FLA*), which profiles the optimization status for the current optimization step. Our designed *FLA* features consist of three parts: 1) Distributional features of the current population, e.g, average distance between any pair of individuals, aiding in assessing the distribution and diversity of candidates; 2) Statistic features of the current objective values, e.g, average objective value gap between an individual and the best-so-far solution, providing insights into the quality of optimization progress; 3) Time stamp embedding, e.g, the number of consumed generations, offering a temporal perspective on the tasks' evolution. Collectively, *FLA* features convey the optimization status of current population on the given task, constantly informing the SEG an inductive bias on generating update rule for next generation. Both the leveraged *VTE* and *FLA* features are task-agnostic and dimension-agnostic, which allows our SYMBOL to generalize across diverse tasks. More details on the state representation and the SEG network architecture can be found in Appendix A.1, and Appendix A.2, respectively.

## 3.2 THREE STRATEGIES FOR TRAINING THE SEG

SYMBOL aims to meta-learn the SEG (parameterized by $\theta$) by optimizing the accumulated meta-objective over a task distribution $\mathbb{D}$, as shown in Eq. (1). Given a BBO problem $f$ sampled from $\mathbb{D}$, in the form of the Markov Decision Process, the SEG receives an optimization status as $state$ and generates a symbolic update rule $\tau^{(t)}$ as $action$ to update the candidate population at each generation $t$. For each generation $t$, we measure how well the generated $\tau^{(t)}$ performs on the sampled $f$ by a meta-performance metric $R(\tau^{(t)}, f)$ as $reward$. We denote the update rules trajectory generated for $f$ as $\Psi_f = \{\tau^{(1)}, \cdots, \tau^{(T)}\}$, where $T$ denotes the maximum number of generations for the lower-level optimization task, and $G(\Psi_f)$ denote the accumulated meta-performance.

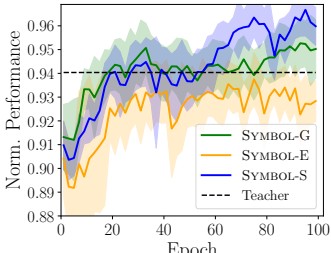

Figure 3: Performance of three different training strategies.

$$J(\theta) = \mathbb{E}_{f \sim \mathbb{D}}\big[G(\Psi_f)|\theta\big] = \mathbb{E}_{f \sim \mathbb{D}}\big[\sum_{t=1}^{T} R(\tau^{(t)}, f)|\theta\big] \tag{1}$$

We introduce three SYMBOL training strategies, catering to diverse application needs. Their effectiveness is illustrated in Figure 3. Each features a distinct reward function $R(\tau^{(t)}, f)$.

**Exploration learning (SYMBOL-E).** Our first proposed strategy SYMBOL-E seeks to directly let the SEG learn to build optimizers from scratch. It utilizes a reward function to measure the effectiveness of the generated optimizer (update rules) based on the progress they facilitate:

$$R_{\text{explore}}(\tau^{(t)}, f) = (-1) \cdot \frac{y^{*,(t)} - y^{\text{opt}}}{y^{*,(0)} - y^{\text{opt}}} \tag{2}$$

where $y^{*,(t)}$ denotes the minimal objective value during the generations $0 \sim t$ and $y^{\text{opt}}$ denotes the known optimal for the given task (considering minimization). However, BBO tasks often present

intricate optimization landscapes, leading to significant exploration hurdles (Reddy et al., 2019). Consequently, when training the SEG using SYMBOL-E, it might demonstrate competitive outcomes but with a relatively slower convergence. This stems from the SEG undergoing numerous trial-and-error attempts before pinpointing proper update rules, as shown by the orange trajectory in Figure 3.

**Guided learning (SYMBOL-G).** Our subsequent strategy, SYMBOL-G, endeavours to train the SEG by mimicking the performance of a leading black-box optimizer $\kappa$ (teacher optimizer). To this end, we let the SEG and the teacher $\kappa$ run in parallel (with identical initial population[1]). For each generation in the lower-level optimization process, we compute the meta-performance $R_{\text{guided}}(\tau^{(t)}, f)$ of SEG by measuring a negation of the Earth Mover's Distance (Rubner et al., 1998) between its current population $x^{(t)}$ and the population of the teacher $x_\kappa^{(t)}$:

$$R_{\text{guided}}(\tau^{(t)}, f) = \frac{-1}{x_{\max} - x_{\min}} \cdot \max\left\{\min\{||x_i^{(t)} - x_{\kappa,j}^{(t)}||_2\}_{j=1}^{|x_\kappa|}\right\}_{i=1}^{|x|} \tag{3}$$

where $x_{\max}$ and $x_{\min}$ denote the upper bound and the lower bound of the search space, $|\cdot|$ denotes the size of the population and $||\cdot||_2$ denotes the Euclidean distance between two individual solutions. Utilizing the demonstration from the teacher optimizer as a guiding policy, SYMBOL-G efficiently addresses the exploration challenges faced in SYMBOL-E, leading to the fastest convergence as shown in Figure 3. Meanwhile, we note that the final performance of SYMBOL-G is on par with or even surpassing the performance of the teacher, underscoring the desirable effectiveness.

**Synergized learning (SYMBOL-S).** Our third strategy SYMBOL-S, integrates both the above designs as shown in Eq. (4) where $\lambda$ is a real-valued weight. Meanwhile, to reduce reliance on known optima (in SYMBOL-E), we introduce a surrogate global optimum, $\hat{y}^{\text{opt}}$, as a dynamic alternative to substitute the static ground truth $y^{\text{opt}}$ in Eq. (2). We refer more details of $\hat{y}^{\text{opt}}$ to Appendix A.4.

$$R_{\text{synergized}}(\cdot) = \hat{R}_{\text{explore}}(\cdot) + \lambda R_{\text{guided}}(\cdot) \tag{4}$$

The results, as captured in Figure 3, reveal that SYMBOL-S excels in synergizing exploration and guided learning, resulting in steadfast convergence and the best final performance.

**Teacher optimizer selection.** Choosing a proper teacher optimizer is generally straightforward, involving selections from proven performers in BBO benchmarks or state-of-the-art domain literature. For training SYMBOL in this paper, we utilize the MadDE (Biswas et al., 2021) as the teacher for SYMBOL-G and SYMBOL-S , since MadDE is one of the winners of the CEC benchmark. Note that our SYMBOL is a teacher-agnostic framework, which is validated in Section 4.3. For scenarios where teacher selection is challenging, SYMBOL-E offers a fallback, allowing for meta-learning an effective policy from scratch, without relying on any teacher optimizer.

**Training algorithm.** Upon selecting one of the aforementioned strategies, SYMBOL can be efficiently trained using reinforcement learning. In our work, we employ the Proximal Policy Optimization with a critic network (Schulman et al., 2017). See pseudocode in Appendix A.5.

## 4 EXPERIMENTAL RESULTS

In this section, we delve into the following research questions: **RQ1**: Does SYMBOL (in any of the three strategies) attain state-of-the-art performance during both meta test and meta generalization? **RQ2**: What insights has SYMBOL gained and how could they be interpreted? **RQ3**: Does the choice of teacher optimizer in SYMBOL-G and SYMBOL-S has significant impact on performance? **RQ4**: How do the hyper-parameters and each component of SYMBOL designs affect the effectiveness? Below, we first introduce the experimental setups and then discuss RQ1 $\sim$ RQ4, respectively.

**Training distribution $\mathbb{D}$.** Our training dataset is synthesized based on the well-known IEEE CEC Numerical Optimization Competition (Mohamed et al., 2021) benchmark, which contains ten challenging synthetic BBO problems ($f_1 \sim f_{10}$). These problems, having a longstanding history in the literature for evaluating the performance of BBO optimizers, show different global optimization

---

[1]In SYMBOL, the student population size is fixed as 100. However, since different teachers may employ different initial population sizes, it may be impossible for the student to share an identical initial population with the teacher. In such situations, we use stratified sampling to form the student's initial population from the teacher's. Details are given in Appendix A.3.

Table 1: Comparisons of SYMBOL and other baselines during meta test and meta generalization.

| | Baselines | Meta Test ($\mathbb{D}$) #Ps=100, #Dim=10, #FEs=50000 | | | Meta Generalization (HPO-B) #Ps=5, #Dim=2~16, #FEs=500 | | | Meta Generalization (Protein-docking) #Ps=10, #Dim=12, #FEs=1000 | | |
|---|---|---|---|---|---|---|---|---|---|---|
| | | Mean↑±(Std) | Time | Rank | Mean↑±(Std) | Time | Rank | Mean↑±(Std) | Time | Rank |
| BBO | RS | 0.932±(0.007) | -/0.03s | 11 | 0.908±(0.004) | -/0.02s | 8 | 0.996±(0.000) | -/0.003s | 4 |
| | MadDE | 0.940±(0.009) | -/0.8s | 6 | 0.932±(0.004) | -/0.2s | 6 | 0.991±(0.001) | -/0.4s | 8 |
| | sep-CMA-ES | 0.935±(0.017) | -/1.3s | 9 | 0.870±(0.017) | -/0.1s | 9 | 0.971±(0.000) | -/0.3s | 10 |
| | ipop-CMA-ES | 0.970±(0.012) | -/1.4s | 3 | 0.938±(0.013) | -/0.1s | 5 | 0.996±(0.003) | -/0.3s | 5 |
| | SMAC | 0.937±(0.019) | -/1.1m | 7 | **0.979**±(0.005) | -/0.7m | 1 | 0.998 ±(0.000) | -/3.8m | 2 |
| MetaBBO | LDE | 0.970±(0.006) | 9h/0.9s | 2 | fail | | | fail | | |
| | DEDDQN | 0.959±(0.007) | 38m/1.1m | 5 | 0.862±(0.026) | -/0.6s | 10 | 0.993±(0.000) | -/1.3s | 7 |
| | Meta-ES | 0.936±(0.012) | 12h/0.4s | 8 | 0.949±(0.002) | -/0.1s | 3 | 0.984±(0.000) | -/0.3s | 9 |
| | MELBA | 0.846±(0.012) | 4h/2.6m | 13 | fail | | | fail | | |
| | RNN-Opt | 0.923±(0.010) | 11h/0.4m | 12 | fail | | | fail | | |
| | SYMBOL-E | 0.934±(0.008) | 6h/0.9s | 10 | 0.920±(0.007) | -/0.5s | 7 | 0.996±(0.000) | -/0.5s | 3 |
| | SYMBOL-G | 0.964±(0.012) | 10h/1.0s | 4 | 0.940±(0.011) | -/0.5s | 4 | 0.995±(0.000) | -/0.5s | 6 |
| | SYMBOL-S | **0.972**±(0.011) | 10h/1.3s | 1 | 0.963±(0.006) | -/0.7s | 2 | **0.999**±(0.000) | -/0.7s | 1 |

Note: For meta generalization, we test on HPO-B and protein-docking tasks featuring different task dimensions (#Dim), population sizes (#Ps), and optimization horizons (#FEs). Note that several MetaBBO methods fail to generalize to these two realistic tasks: RNN-Opt and MELBA are not generalizable across different task dimensions; LDE is not generalizable across different population sizes.

properties such as uni-modal, multi-modal, (non-)separable, and (a)symmetrical features. Additionally, they reveal varied local landscape properties, such as different properties around different local optima, continuous everywhere yet differentiable nowhere landscapes, and optima situated in flattened areas. To augment those 10 functions to form a boundless training data distribution for reinforcement learning, we adopt a sampling procedure: 1) setting a random seed; 2) randomly select a batch of $N$ problems from $f_1 \sim f_{10}$ (with allowance for repetition), where we set problem dimension to 10 and the searching space to $[-100, 100]^{10}$; 3) for each selected problem $f$, randomly introduce a offset $z \sim U[-80, 80]^{10}$ to the optimal and then randomly generate a rotational matrix $M \in \mathbb{R}^{10 \times 10}$ to rotate the searching space; and 4) yield the transformed problems $f(M^{\mathrm{T}}(x + z))$.

**Baselines.** We compare SYMBOL with a wide range of traditional BBO and MetaBBO optimizers. Regarding the BBO methods, we include **Random Search (RS)**, **MadDE** (Biswas et al., 2021), strong ES optimizers **sep-CMA-ES** (Ros & Hansen, 2008) and **ipop-CMA-ES** (Auger & Hansen, 2005), and a strong BO optimizer **SMAC** (Lindauer et al., 2022). Regarding the MetaBBO methods, we include **DEDDQN** (Sharma et al., 2019), **LDE** (Sun et al., 2021), and **Meta-ES** (Lange et al., 2022) as baselines for auto-configuration; **RNN-Opt** (TV et al., 2019) and **MELBA** (Chaybouti et al., 2022) as baselines for directly generating the next candidate solutions. All hyper-parameters were tuned using the grid search and we list their best settings in Appendix B.1.

**Training & code release.** We dispatch training details and hyper-parameter setups of our SYMBOL in Appendix B.2. For MetaBBO baselines, they are all trained on the same problem distribution $\mathbb{D}$ as our framework while following their recommended settings. We release the implementation python codes at https://github.com/GMC-DRL/Symbol, where we show how to train SYMBOL with different strategies, and how to generalize it to unseen problems.

## 4.1 PERFORMANCE EVALUATION (RQ1)

To ensure a thorough assessment, we conduct two sets of experiments: meta test and meta generalization. The former evaluates the performance of the trained optimizer on unseen, yet in-distribution (w.r.t. $\mathbb{D}$) BBO problems. The latter evaluates the adaptability of the optimizer on completely novel realistic BBO tasks, characterized by varying task dimensions (#Dim), population sizes (#Ps), and optimization horizons (#FEs). For each method, we report in Table 1: 1) the average of min-max normalized performance with the corresponding standard deviation over 5 independent runs (see Appendix B.3 for normalization details); 2) overall training time and the average solving time per instance, in the format of "training/testing time"; and 3) performance rank among all methods.

**Meta test.** We meta-test SYMBOL (trained with the three strategies) on a group of 320 unseen problem instances sampled from $\mathbb{D}$, and compare them with all baselines. For each instance, all participants are allowed to optimize it for no more than $5 \times 10^4$ function evaluations (FEs). We note that SMAC is an exception which is given 500 FEs, otherwise it takes several days for evaluating under the same FEs. The results in Table 1 show that: 1) SYMBOL-S stands out by delivering a performance that not only surpasses all the baselines but also significantly outpaces its teacher, MadDE; 2) Compared with SYMBOL-G which concentrates itself on the teacher demonstration and SYMBOL-E which probes optimal update rules from scratch, SYMBOL-S attain a significant improvement by

Table 2: Generated update rules and corresponding frequencies.

| Update rule $(x + \underline{\tau})$ | Frequency |
|---|---|
| $x + 0.18 \times (x^* - x_r) + 0.42 \times (x_i^* - x_r)$ | 42.207% |
| $x + \overline{0.18 \times (x^* - x) + 0.42 \times (x_i^* - x)}$ | 39.448% |
| $x + \overline{0.18 \times (x^* - x_i^*)}$ | 7.448% |
| $x + 0.6 \times (x^* - x_r) + \overline{0.6 \times (x_i^* - x)} + 0.18 \times (x^* - x_i^*)$ | 2.601% |
| $x + \overline{0.78 \times (x^* - x_r) + 0.42 \times (x_i^* - x_r)}$ | 1.586% |

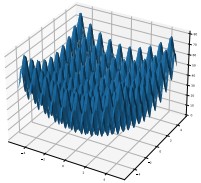

Table 3: 2D Rastrigin.

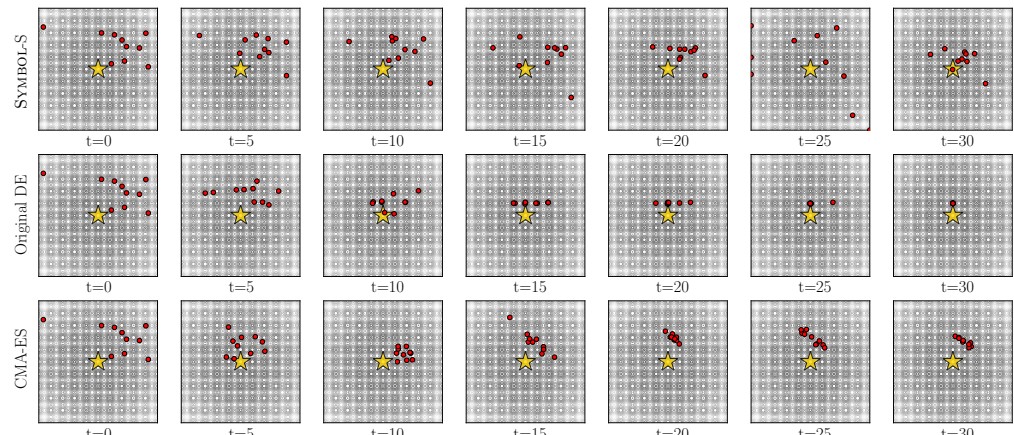

Figure 4: Evolution visualization of the optimizers, showing the position of population (red dots) and the global optimal (yellow star). **Top**: SYMBOL-S; **Middle**: original DE; **Bottom**: CMA-ES.

integrating their meta-objectives coordinately; 3) Methods like RNN-Opt and MELBA, which employ coordinate-based features, may be prone to overfitting, thereby resulting in suboptimal generalization to unseen instances, even if they are from the same distribution, $\mathbb{D}$. In contrast, our SYMBOL leverages generalizable VTE and VLA features and achieves the best performance.

**Meta generalization.** To test the zero-shot generalization performance on realistic tasks, we consider the hyper-parameter optimization benchmark HPO-B (Arango et al., 2021) and the Protein-docking benchmark (Hwang et al., 2010), details of which are elaborated in Appendix B.5. The results in Table 1 show that: 1) SYMBOL-S not only achieves the state-of-the-art zero-shot generalization performance against all MetaBBO baselines, it also achieves similar performance with the advanced BO tool-box SMAC; 2) While SMAC, developed over decades, takes 42s and 227s for HPO and Protein-docking respectively, the optimizers discovered by our SYMBOL are able to achieve comparable results with less than 1s, underscoring the significance of our SYMBOL framework; 3) Recent MetaBBO methods DEDDQN and Meta-ES could be severely disturbed by the intricate landscape properties (even worse than exhaustive random search). Instead, SYMBOL with the proposed three strategies present a more robust zero-shot generalization performance.

## 4.2 INTERPRETING SYMBOL (RQ2)

We now examine and interpret the generated closed-form update rules by our SYMBOL-S. Using a 2D Rastrigin function (Figure 3), known for its challenging multi-modal nature, we compare the performance of the trained SYMBOL-S with original DE (Storn & Price, 1997) and CMA-ES (Hansen et al., 2003). Each optimizer evolves a population of 10 solutions over 30 generations. SYMBOL-S generates a variety of update rules and we detail the top-5 rules based on 50 runs in Table 2. Notably, the two most frequent rules showcase distinct optimization strategies: the first emphasizes exploration in uncharted territories using $x_r$, while the second prioritizes convergence towards previously identified optimal solutions using the current $x$. In Figure 4, we illustrate the search behaviors of our learned SYMBOL-S compared to the baseline. The visualization clearly reveals that SYMBOL-S is able to avoid falling into local optima and successfully locate the global optima by dynamically deciding a proper update rule, while DE and CMA-ES show premature due to their inflexible manual update rules. We validate this conclusion by further investigating the update rules

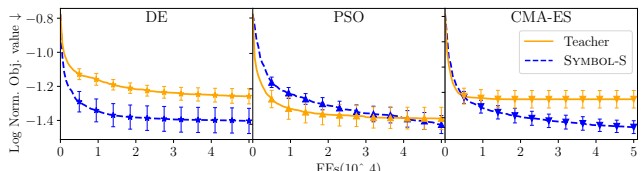

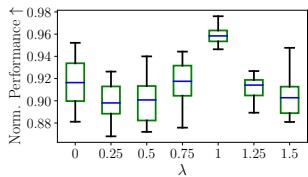

Figure 5: Performance of SYMBOL-S with different teachers.    Figure 6: Effects of different $\lambda$.

generated in different periods of the optimization steps. In the later stage of the showcased evolution path ($20 \leq t \leq 30$), our SYMBOL-S (the top row) shows two searching patterns: 1) during $20 \leq t \leq 25$, the population diverges to undiscovered area. We look back at the corresponding generated update rules in this period and find it is $0.18 \times (x^* - x_r) + 0.42 \times (x_i^* - x_r)$, which introduces $x_r$ for exploring potential optimal area; 2) during $25 \leq t \leq 30$, it prompts the population converging to some local area. The dominant updated rule in this period is $0.18 \times (x^* - x) + 0.42 \times (x_i^* - x)$, accelerating the population to converge to the most likely optima.

### 4.3 TEACHER OPTIONS (RQ3)

We now investigate more teacher options where we substituted MadDE with the original DE, PSO, and CMA-ES, respectively. Figure 5 presents the log-normalized, step-by-step objective values for these variants in comparison to their teacher optimizers. Impressively, SYMBOL-S not only follows the guidance of its teacher optimizer but consistently surpasses its performance during training. This demonstrates the adaptability of our framework, capable of leveraging a wide array of existing and future BBO optimizers as teachers. Comprehensive results, including those of SYMBOL-S on HPO-B and Protein-docking and the outcomes for SYMBOL-G, are detailed in Figure 8, Appendix C.1.

### 4.4 SENSITIVITY ANALYSIS AND ABLATION STUDIES (RQ4)

**Effects of $\lambda$.** In Figure 6, we report the normalized performance (5 runs) of different $\lambda$ values on HPO-B tasks. When $\lambda$ is set too low, behavior cloning may be ineffective, resulting in SYMBOL-S performing similarly to SYMBOL-E. On the other hand, a high value of $\lambda$ can potentially cause overfitting. Together with the results on Protein-docking which we leave at Appendix C.2, Figure 10, we suggest setting $\lambda = 1$ to strike a good balance between the imitation and the exploration.

**Abalation on FLA features and basis symbol set.** To verify that the FLA features help SYMBOL attain good generalization performance, we conduct ablation studies on SYMBOL-S in Appendix C.2. We find that each element of the FLA features plays a crucial role in ensuring robust generalization capabilities. Besides the FLA features, the choice of tokens in the basis symbol set $\mathbb{S}$ may also affect the performance, as discussed in Appendix C.2. We find that the key to selecting the basis symbol set is ensuring its alignment with the BBO domain, rather than simply increasing the token count.

## 5 CONCLUSIONS

In this paper, we introduce SYMBOL, a novel MetaBBO framework tailored to autonomously generate optimizers for BBO tasks. Adopting a bi-level structure, SYMBOL integrates an SEG network at the lower level, adept at generating closed-form BBO update rules. The SEG is then meta-learned via three distinct training strategies: SYMBOL-E, SYMBOL-G, and SYMBOL-S, which possesses the versatility to learn update rules either from scratch and/or by leveraging insights from established BBO optimizers. Experimental results underscore the effectiveness of SYMBOL: training on synthetic tasks suffices for it to generate optimizers that surpass state-of-the-art BBO and MetaBBO methods, even when applied to entirely unseen realistic BBO tasks. As a pioneer work on generative MetaBBO through symbolic equation learning, while our SYMBOL offers several advancements, certain limitations exist. Selecting a teacher for new BBO tasks can be challenging: even though SYMBOL-E provides a teacher-free approach, future work may further enhance its training sample efficiency (as faced by other RL tasks). Additionally, exploration of more powerful symbol sets is another important future work. Furthermore, researching more advanced generators, such as those based on large-language models (LLMs) (OpenAI, 2023), is also a promising future direction.

ACKNOWLEDGMENTS

This work was supported in part by the National Natural Science Foundation of China under Grant 62276100, in part by the Guangdong Natural Science Funds for Distinguished Young Scholars under Grant 2022B1515020049, in part by the Guangdong Regional Joint Funds for Basic and Applied Research under Grant 2021B1515120078, in part by the TCL Young Scholars Program, and in part by the Singapore MOE AcRF Tier 1 funding (RG13/23).

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

## A    TECHNICAL DETAILS OF SYMBOL

### A.1    CONTEXTUAL STATE REPRESENTATION

**VTE.** We first represent each token in $\mathbb{S}$ with a 4-bit binary code, e.g., 0001 for $+$, 0010 for $-$, etc., listed in in Table 4. During constructing the update rule $\tau$, we embed the corresponding under-constructed symbolic expression tree into a fixed-length vector which is called VTE as a contextual state to inform the SEG prompting next token for $\tau$. We showcase a symbolic expression tree with maximum height $H = 3$ in Figure 7. In this case, the SEG has generated 4 tokens in the previous steps ($+$, $x^*$, $\times$, $c$), thus the VTE for the current half-constructed tree is shown in the bottom of Figure 7. For those blank positions in this tree we use 0000 to represent them, while for those positions holding by a token $\tau_i$, we use the binary codes in Table 4 to represent them. As we have set a maximum height of $\tau$ to 5 in this paper, the VTE embedding of the symbolic expression tree $\tau$ can be represented by 31 4-bit binary codes, resulting in a vector of $\{0, 1\}^{124}$.

**FLA.** The formulation of the *FLA* features is described in Table 5. States $s_{\{1,2,3\}}$ collectively represent the distributional features of the current candidate population. Specifically, state $s_1$ represents the average distance between each pair of candidate solutions, indicating the overall dispersion level. State $s_2$ represents the average distance between the best candidate solution in the current population and the remaining solutions, providing insights into the convergence situation. State $s_3$ represents the average distance between the best solution found so far and the remaining solutions, indicating the exploration-exploitation stage. Then, States $s_{\{4,5,6\}}$ provide information about the statistics of the objective values in the current population and contribute to our framework's understanding of the fitness landscape. Specifically, state $s_4$ represents the average difference between the best objective value found in the current population and the remaining solutions, and $s_5$ represents the average difference when compared with the best objective value found so far. State $s_6$ represents the standard deviation of the objective values of the current candidates. Finally, states $s_{\{7,8,9\}}$ collectively represent the time-stamp features of the current optimization process. Among them, state $s_7$ denotes the current process, which can inform the framework about when to adopt appropriate strategies. States $s_8$ and $s_9$ are measures for the stagnation situation.

Table 4: Binary encoding of tokens in $\mathbb{S}$.

| Tokens | Binary code | Tokens | Binary code |
|--------|-------------|--------|-------------|
| $+$ | 0001 | $\times$ | 0010 |
| $-$ | 0011 | $c$ | 0100 |
| $x$ | 0101 | $x^*$ | 0110 |
| $x_-$ | 0111 | $\Delta x$ | 1000 |
| $x_r$ | 1001 | $x_i^*$ | 1010 |

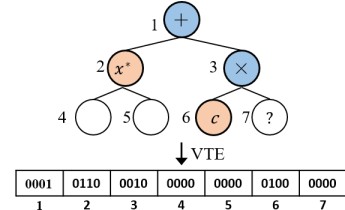

Figure 7: Illustration of VTE transformation.

### A.2    NETWORK ARCHITECTURE DETAILS

**SEG (Actor Network).** The network parameters of SEG $\theta$ includes three components: 1) FLA Embedder with parameters $\{W_{\text{FLA}}, b_{\text{FLA}}\}$; 2) LSTM with parameters $\{W_h, b_h, W_{in}, b_{in}\}$; 3) Constant inference decoder with parameters $\{W_\varpi, b_\varpi, W_\epsilon, b_\epsilon\}$. To generate an update rule $\tau$ for the current optimization step (generation $t$), we first embed the FLA features of the current optimization status $S_{\text{FLA}}$ as the initial cell state:

$$c_{\text{FLA}}^0 = W_{\text{FLA}}^{\text{T}} S_{\text{FLA}} + b_{\text{FLA}} \tag{5}$$

We then initialize the $h^{(0)} = \mathbf{0}$ and VTE($\tau$) (at the very beginning, $\tau$ is empty) as the other two contextual states. The LSTM then starts auto-regressively determining each $\tau_i$ by iteratively executing the following three steps:

$$h^{(i)}, c_{\text{FLA}}^{(i)}, o^{(i)} = \text{LSTM}(\text{VTE}(\tau), c_{\text{FLA}}^{(i-1)}, h^{(i-1)}) \tag{6}$$

$$\tau_i \sim \text{Softmax}(o^{(i)}) \tag{7}$$

$$\tau = \tau \cup \{\tau_i\} \tag{8}$$

Table 5: Formulations of FLA features.

| | | States | Notes |
|---|---|---|---|
| **Population** | $s_1$ | $mean_{i,j}\|\|x_i^{(t)} - x_j^{(t)}\|\|_2$ | Average distance between any pair of individuals in current population. |
| | $s_2$ | $mean_i\|\|x_i^{(t)} - x^{*,(t)}\|\|_2$ | Average distance between each individual and the best individual in $t^{\text{th}}$ generation. |
| | $s_3$ | $mean_i\|\|x_i^{(t)} - x^*\|\|_2$ | Average distance between each individual and the best-so-far solution. |
| **Objective** | $s_4$ | $mean_i(f(x_i^{(t)}) - f(x^*))$ | Average objective value gap between each individual and the best-so-far solution. |
| | $s_5$ | $mean_i(f(x_i^{(t)}) - f(x^{*,(t)}))$ | Average objective value gap between each individual and the best individual in $t^{th}$ generation. |
| | $s_6$ | $std_i(f(x_i^{(t)}))$ | Standard deviation of the objective values of population in $t^{th}$ generation, a value equals 0 denotes converged. |
| **Time Stamp** | $s_7$ | $(T - t)/T$ | The potion of remaining generations, $T$ denotes maximum generations for one run. |
| | $s_8$ | $st/T$ | $st$ denotes how many generations the optimizer stagnates improving. |
| | $s_9$ | $\begin{cases} 1 & \text{if } f(x^{*,(t)}) < f(x^*) \\ 0 & \text{otherwise} \end{cases}$ | Whether the optimizer finds better individual than the best-so-far solution. |

If the constant token $c$ is inferred at $i^{th}$ construction step ($\tau_i = c$), we activate the constant inference decoder to infer the concrete constant value:

$$\varpi \sim \text{Softmax}(W_\varpi^\text{T} o^{(i)} + b_\varpi) \tag{9}$$

$$\epsilon \sim \text{Softmax}(W_\epsilon^\text{T} o^{(i)} + b_\epsilon) \tag{10}$$

$$c = \varpi \times 10^\epsilon \tag{11}$$

**Critic Network.** The Critic has parameters $W_{\text{critic}}, b_{\text{critic}}$. It receives the FLA features as the input and predicts the expected return value $\upsilon$ (baseline value) as follows:

$$\upsilon = W_{\text{critic}}^\text{T} S_{\text{FLA}} + b_{\text{critic}} \tag{12}$$

### A.3 POPULATION SAMPLING

When SYMBOL-G or SYMBOL-S constructs its initial population during the training process, we need to ensure SYMBOL have the same initial population with the teacher optimizer. However, considering that the population size of SYMBOL may be different from the teacher optimizer, and some teacher optimizers even dynamically decay the population size through the optimization process, a flexible method is needed to initialize SYMBOL's population. This method should not only adapt to the uncertain population size of teacher but also make the initial population of SYMBOL closely resemble that of the teacher optimizer. Specifically, in the case where the teacher's population size is larger than that of SYMBOL, the population of SYMBOL is stratified sampled from the teacher optimizer base on the objective value. In cases where teacher's population size is smaller than SYMBOL, SYMBOL firstly replicates the teacher's population and then randomly samples candidates from the search space until the required population size is reached. We illustrate the population sampling method in Algorithm 1.

### A.4 SURROGATE GLOBAL OPTIMUM

The teacher optimizer in SYMBOL-S provides a convenience to reduce the reliance on known optima in SYMBOL-E. We propose a surrogate global optimum, $\hat{y}^{\text{opt}}$, to substitute the ground true global

---

**Algorithm 1** Sampling student's initial population from teacher's population.

---

**input:** Teacher's population $p^{\text{tea}}$; Teacher's population size $Ps^{\text{tea}}$; Student's population size $Ps^{\text{stu}}_*$
**output:** Student's population $p^{\text{stu}}$
1:   $p^{\text{stu}} \leftarrow \emptyset$; $Ps^{\text{stu}} \leftarrow 0$
2:   **if** $Ps^{\text{stu}}_* > Ps^{\text{tea}}$ **then**
3:      $p^{\text{stu}} \leftarrow p^{\text{stu}} \cup p^{\text{tea}}$; $Ps^{\text{stu}} \leftarrow Ps^{\text{tea}}$
4:      **while** $Ps^{\text{stu}} < Ps^{\text{stu}}_*$ **do**
5:         Randomly sample a solution $x_r$ from the searching space
6:         $p^{\text{stu}} \leftarrow p^{\text{stu}} \cup \{x_r\}$
7:         $Ps^{\text{stu}} \leftarrow Ps^{\text{stu}} + 1$
8:      **end while**
9:   **else**
10:      Sort $p^{\text{tea}}$ upon the objective value
11:      Equally spaced sample $\#Ps^{\text{stu}}_*$ candidate solutions from $p^{\text{tea}}$, store in $p^{\text{sample}}$
12:      $p^{\text{stu}} \leftarrow p^{\text{sample}}$; $Ps^{\text{stu}} \leftarrow Ps^{\text{stu}}_*$
13: **end if**

---

optimum. In the very beginning of training process, $\hat{y}^{\text{opt}}$ is determined from the teacher optimizer when it optimizes the corresponding training task. In the rest of training process, $\hat{y}^{\text{opt}}$ is dynamically updated when either the teacher or the SEG finds a solution with lowest objective value thus far.

### A.5 PSEUDO CODE FOR THE TRAINING PROCESS

Algorithm 2 illustrates the pseudo code for the training process of three strategies of SYMBOL. To execute these strategies, one must have access to the training distribution $\mathbb{D}$ and set the learning rate $\alpha$. Additionally, for SYMBOL-G and SYMBOL-S, the teacher optimizer $\kappa$ is required. First, we initialize the SEG with LSTM parameters $\theta$. In each meta-training step, a batch of problems $f$ are sampled from $\mathbb{D}$. Then, we reset $\kappa$ for SYMBOL-G and SYMBOL-S, initialize its population and evaluate the solutions. The population for student is similarly initialized. For each step in lower-level optimization, we employ $\kappa$ to gain the teacher suggestion, which will be skipped when the training strategy is SYMBOL-E. Next, for all training strategies, we generate the update rule for the student population with our SEG and use the rule to advance the population. The reward will be then calculated according to different training modes as formulated in Eq. (2) $\sim$ (4). After the lower-level loop concludes, we update the meta-objective and token trajectory, which participant in the gradient calculation in PPO manner. Upon the meta-training loop, the well-trained SEG with LSTM parameters $\theta$ is returned.

## B   TRAINING AND TEST SETUP

### B.1   SETTINGS OF TRADITIONAL BASELINES

For the sake of comparison fairness, we conduct grid search on some of the hyper-parameters of baselines on the training distribution $\mathbb{D}$. In this section, we present the search results of two BBO baselines, MadDE and sep-CMA-ES, as they are sensitive to hyper-parameter values. For MadDE, we focus on the following hyper-parameters: the probability of qBX crossover ($P_{qBX}$), the percentage of population in p-best mutation ($p$), the initial values for scale factor memory ($F_0$) and crossover rate memory ($CR_0$). For sep-CMA-ES, we consider the following hyper-parameters: the initial center ($\mu_0$) and initial standard deviation ($\sigma_0$). The search range and best settings after gird search of the two baseline algorithms are listed in Table 6. In the experiments of this paper, for MadDE, we choose $P_{qBX}$ of $0.01$ , $p$ of $0.18$, $F_0$ of $0.2$, and $CR_0$ of $0.2$. For sep-CMA-ES, we use $\mu_0$ with a vector of **3** and $\sigma_0$ with a vector of **2**.

### B.2   SETTINGS OF SYMBOL

The neural parameters of our framework consist of two parts: 1) The Actor   (SEG) includes an FLA embedder for embedding the FLA features into the cell state of LSTM, which has parameters

---

**Algorithm 2** Meta-learning the SEG with policy gradient.

---

**input:** Training distribution $\mathbb{D}$; Teacher optimizer $\kappa$; Learning rate $\alpha$; Training strategy;
**output:** Best performing SEG

1: Initialize the SEG with LSTM parameters $\theta$
2: *#Meta-level Looping*
3: **repeat**
4:     Sample problem $f$ from $\mathbb{D}$                 ▷ See Training distribution part in Section 3.2
5:     Reset teacher $\kappa$                       ▷ Skipped if performing SYMBOL-E
6:     Meta-objective $G \leftarrow 0$; Token trajectory $\Psi \leftarrow \emptyset$
7:     Initialize teacher population $X_t, Y_t = f(X_t)$         ▷ Skipped if performing SYMBOL-E
8:     Initialize student population $X_s \leftarrow \begin{cases} X_t, & \text{If SYMBOL-G / S} \text{ ▷ Share the same distribution} \\ X_s, & \text{If SYMBOL-E} \end{cases}$
9:     $Y_s \leftarrow f(X_s)$                         ▷ Evaluate the candidate solutions
10:     *#Lower-level Looping*
11:     **repeat**
12:         $X_t, Y_t \leftarrow \kappa(X_t, Y_t)$          ▷ Teacher's round, skipped if performing SYMBOL-E
13:         $\tau, p(\tau|\theta) \leftarrow SEG_\theta(X_s, Y_s)$       ▷ Generate update rule for student population
14:         $X_s \leftarrow X_s + \tau, Y_s \leftarrow f(X_s)$       ▷ Student's round to suggest new solutions
15:         $R \leftarrow \begin{cases} R_{\text{explore}}(\cdot, f), & \text{If Training strategy is SYMBOL-E} \\ R_{\text{guided}}(\cdot, f), & \text{If Training strategy is SYMBOL-G} \\ R_{\text{synergized}}(\cdot, f), & \text{If Training strategy is SYMBOL-S} \end{cases}$
16:         $G \leftarrow G + R$                      ▷ Accumulate rewards
17:         $\Psi \leftarrow \Psi \cup \{\tau\}$
18:     **until** Function evaluations run out
19:     $\hat{g} \leftarrow G \cdot \nabla_\theta \log\ p(\Psi|\theta)$           ▷ Compute policy gradients
20:     $\theta \leftarrow \theta + \alpha\hat{g}$                  ▷ Apply policy gradients
21: **until** Exceed the pre-defined learning steps

---

Table 6: Grid search for the hyper-parameters of MadDE and sep-CMA-ES.

| Baselines | Parameters | Search range | Chosen values |
|-----------|-----------|--------------|---------------|
| MadDE | $P_{qBX}$ | $[0.01, 0.05, 0.1, 0.5]$ | 0.01 |
| | $p$ | $[0.09, 0.18, 0.27, 0.36]$ | 0.18 |
| | $F_0$ | $[0.1, 0.2, 0.3, 0.4]$ | 0.2 |
| | $CR_0$ | $[0.1, 0.2, 0.3, 0.4]$ | 0.2 |
| sep-CMA-ES | $\mu_0$ | **[0, 1, 2, 3]** | **3** |
| | $\sigma_0$ | **[0, 1, 2, 3]** | **2** |

$\{W_{\text{FLA}} \in \mathbb{R}^{9 \times 16}, b_{\text{FLA}} \in \mathbb{R}^{16}\}$. The LSTM network for generating symbol distribution has parameters $\{W_h \in \mathbb{R}^{16 \times 64}, b_h \in \mathbb{R}^{64}, W_{\text{in}} \in \mathbb{R}^{124 \times 64}, b_{\text{in}} \in \mathbb{R}^{64}\}$, along with two feed-forward layers for inferring the constants, which have parameters $\{W_\epsilon \in \mathbb{R}^{16 \times 2}, b_\epsilon \in \mathbb{R}^2\}$ and $\{W_\varpi \in \mathbb{R}^{16 \times 21}, b_\varpi \in \mathbb{R}^{21}\}$. 2) The Critic is a feed-forward layer with parameters $\{W_{\text{critic}} \in \mathbb{R}^{9 \times 1}, b_{\text{critic}} \in \mathbb{R}^1\}$. Our framework is trained by the PPO algorithm, which updates the parameters $k = 3$ times every $n = 10$ sampling step, with a decay parameter $\gamma = 0.9$. The learning rate $\alpha$ is $10^{-3}$. The number of generations ($T$) for lower-level optimization is 500. The tunable parameter $\lambda$ for SYMBOL-S is set to 1. We simultaneously sample a batch of $N = 32$ problems from problem distribution $\mathbb{D}$ for training. The pre-defined maximum learning steps for PPO is $5 \times 10^4$. Once the training is done, SYMBOL can be directly applied to meta-test (unseen problems sampled from $\mathbb{D}$) and meta-generalization (unseen tasks out of $\mathbb{D}$) without the need for teacher optimizer or further fine-tuning. All experiments are run on a machine with Intel i9-10980XE CPU, RTX 3090 GPU and 32GB RAM.

### B.3 NORMALIZATION OF THE OBJECTIVE VALUES

Considering minimization problem, we first record the worst objective $f_{\max}^k$ and the best objective $f_{\min}^k$ in all runs for the $k^{\text{th}}$ instance ($f_{\min}^k$ is the $f^{\text{opt}}$ if the optima is known). For the $m^{\text{th}}$ run on the $k^{\text{th}}$ task instance, we then record the best-found objective value $f_{\min}^{m,k}$. Then, the min-max normalized

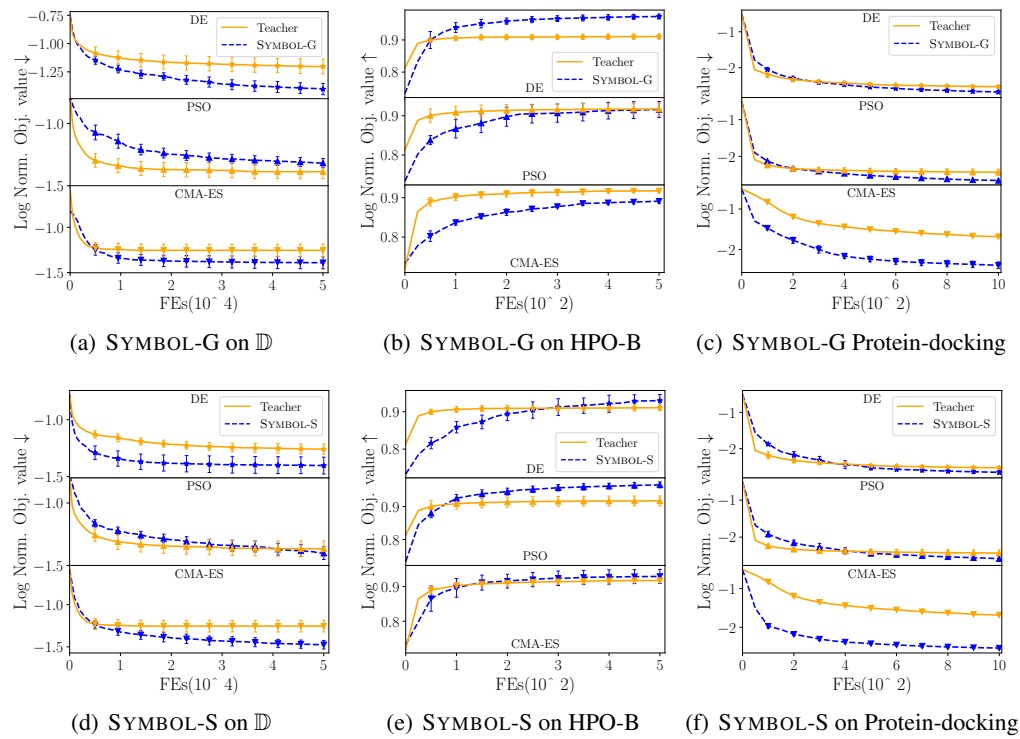

Figure 8: The performance of SYMBOL-G, SYMBOL-S and their teachers on the three benchmark sets: $\mathbb{D}$, HPO-B and Protein-docking.

objective value $Obj$ for a baseline is computed as $\frac{1}{K}\sum_{k=1}^{K}\left[\frac{1}{M}\sum_{m=1}^{M}\left(\frac{f_{\min}^{m,k}-f_{\min}^{k}}{f_{\max}^{k}-f_{\min}^{k}}\right)\right]$, where $K$ and $M$ is the size of test set and the total test runs respectively. The corresponding min-max normalized performance and log-normalized objective value used in our experimental results are computed as $(1-Obj)$ and $\log_{10}(Obj)$.

### B.4 MORE DETAILS OF TRAINING DISTRIBUTION

Table 7 provides more detailed properties of the ten classes of functions in $\mathbb{D}$. These problems include various optimization properties such as unimodal, multi-modal, non-separable, asymmetrical, diverse local or global optimization properties and so on. Furthermore, we also visualize these functions in 2D configuration in Figure 9 for more intuitive insight. By covering such a wide range of landscape features and incorporating fitness landscape analysis based state representation, SYMBOL is not learning how to solve a given task, instead, it learns how to generate different update patterns facing with various landscape feature, which ensures its superior generalization performance. Additionally, for each selected function, we further apply the random offset and rotating operation to augment the function. This augmentation further enrich the training distribution and may enhance the generalization ability of SYMBOL to unseen tasks.

### B.5 TASKS SETS FOR META GENERALIZATION

We provide two challenging realistic task sets to evaluate the zero-shot generalization performance of SYMBOL and other baselines:

HPO-B benchmark (Arango et al., 2021), which includes a wide range of hyperparameter optimization tasks for 16 different model types (e.g., SVM, XGBoost, etc.). These models have various search spaces ranging from $[0,1]^2$ to $[0,1]^{16}$. Each model is evaluated on several datasets, resulting in a total of 86 tasks. For each instance, all baselines are allowed to optimize within 500 function evaluations and with a population size of 5 (for baselines that incorporate a population of candidate

Table 7: Summary of the ten classes of functions in $\mathbb{D}$.

| | No. | Functions | Properties |
|---|---|---|---|
| Unimodal Function | 1 | Bent Cigar Function | -Unimodal
-Non-separable
-Smooth but narrow ridge |
| Basic Functions | 2 | Schwefel's Function | -Multi-modal
-Non-separable
-Local optima's number is huge |
| | 3 | bi-Rastrigin Function | -Multi-modal
-Non-separable
-Asymmetrical
-Continuous everywhere yet differentiable onwhere |
| | 4 | Rosenbrock's plus Griewangk's Function | -Non-separable
-Optimal point locates in flat area |
| Hybrid Functions | 5 | Hybrid Function 1 ($N = 3$) | -Multi-modal or Unimodal, depending on the basic function |
| | 6 | Hybrid Function 2 ($N = 4$) | -Non separable subcomponents |
| | 7 | Hybrid Function 3 ($N = 5$) | -Different properties for different variables subcomponents |
| Composition Functions | 8 | Composition Function 1 ($N = 3$) | -Multi-modal |
| | 9 | Composition Function 2 ($N = 4$) | -Non-separable -Asymmetrical |
| | 10 | Composition Function 3 ($N = 5$) | -Different propeties arround different local optima |
| Search range: $[-100, 100]^D$ | | | |

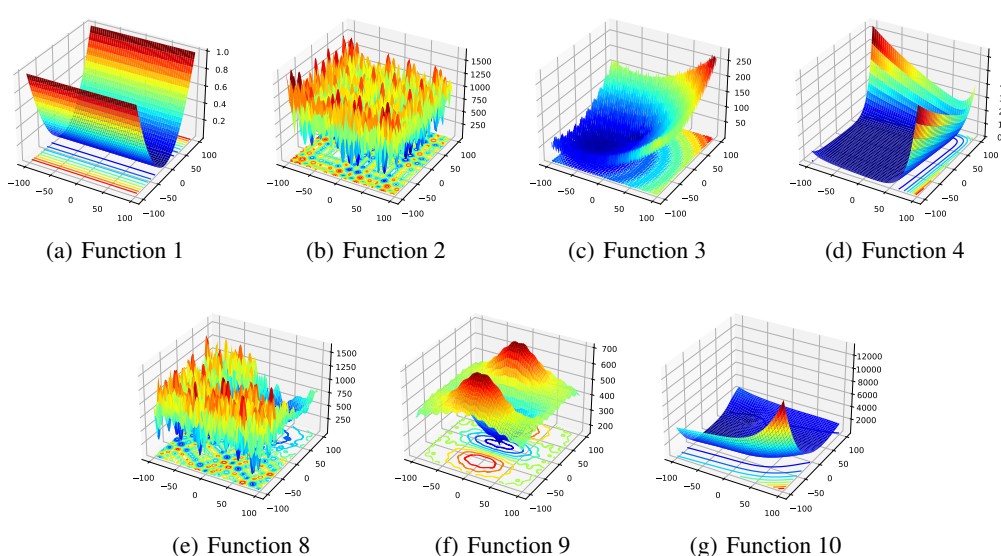

(a) Function 1     (b) Function 2     (c) Function 3     (d) Function 4

(e) Function 8     (f) Function 9     (g) Function 10

Figure 9: Fitness landscapes of functions in $\mathbb{D}$ when dimension is set to 2. Functions 5-7 can not be visualized in the 2D configuration due to the inherent constraint from hybrid functions.

solutions). To save evaluation time, we adopt the continuous version of HPO-B, which provides surrogate evaluation functions for time-consuming machine learning tasks.

Protein-docking benchmark (Hwang et al., 2010), where the objective is to minimize the Gibbs free energy resulting from protein-protein interaction between a given complex and any other conformation. We select 28 protein complexes and randomly initialize 10 starting points for each complex. All 280 task instances have a search space of $[-100, 100]^{12}$. For each instance, all baselines are allowed to optimize within 1000 function evaluations and with a population size of 10.

Table 8: Ablation studies on the feature selection in FLA features.

| Task | SYMBOL-S | SYMBOL-S w/o pop | SYMBOL-S w/o obj | SYMBOL-S w/o ts | SYMBOL-S with xy |
|---|---|---|---|---|---|
| CEC-competition↑ | **0.972**±(0.011) | 0.926±(0.015) | 0.964±(0.011) | 0.910±(0.017) | 0.849±(0.012) |
| HPO-B↑ | **0.963**±(0.006) | 0.937±(0.009) | 0.875±(0.027) | 0.779±(0.024) | fail |
| Protein-docking↑ | **0.999**±(0.000) | 0.989±(0.002) | 0.996±(0.001) | 0.991±(0.001) | fail |

Table 9: Ablation study results for symbols set design.

| Task | SYMBOL-S | Symbols set | |
|---|---|---|---|
| | | SYMBOL-S with $\mathbb{S}^+$ | SYMBOL-S with $\mathbb{S}^-$ |
| CEC-competition↑ | **0.972**±(0.011) | 0.952±(0.015) | 0.930±(0.018) |
| HPO-B↑ | **0.963**±(0.006) | 0.937±(0.009) | 0.910±(0.009) |
| Protein-docking↑ | **0.999**±(0.000) | 0.995±(0.001) | 0.990±(0.002) |

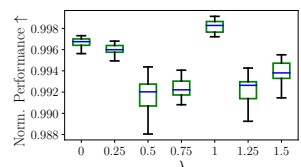

Figure 10: Performance with different $\lambda$ on Protein-docking.

## C  ADDITIONAL EXPERIMENT RESULTS

### C.1  TEACHER OPTIONS

Additionally, we provide a comprehensive analysis of the potential teacher options for SYMBOL-G and SYMBOL-S. By incorporating different teachers (as described in the main body of this paper), we train the resulting variants on $\mathbb{D}$. The meta-test results and meta-generalization results are illustrated in Figure 8. The results are consistent with the observations made in Section 4.3. SYMBOL-S achieves robust performance improvement, while SYMBOL-G fails to learn from PSO. This issue may be attributed to the definitive update rules, which can lead to overfitting. Even for SYMBOL-S, where we integrate exploration behavior and prior demonstrations, our framework encounters difficulties in learning from this type of teacher. We acknowledge this as a potential area for future work and aim to address this limitation.

### C.2  SENSITIVITY ANALYSIS AND ABLATION STUDIES

**Tunable parameter $\lambda$.** We report the normalized zero-shot performance (5 runs) of SYMBOL-S trained with 7 different $\lambda = \{0, 0.25, 0.5, 0.75, 1, 1.25, 1.5\}$ on the Protein-docking tasks in Figure 10. The results are basically consistent with the observation in Section 4.4. We found that $\lambda = 1$ is a relatively better choice.

**Feature selection for state representation.** To verify that the FLA features help SYMBOL attain good generalization performance, we conduct ablation studies on SYMBOL-S by omitting various components: the distributional features of the current population (denoted as w/o pop), the statistic features of the current objective values (w/o obj), and the time stamp embedding (w/o ts). Additionally, we also assessed the performance when replacing FLA features with the simple coordinate-based feature set $\{x, f(x)\}$ (with xy), which follows the coordinate state representation design in (Chen et al., 2017; TV et al., 2019; Chaybouti et al., 2022). The outcomes, presented in Table 8, confirm that each element of the FLA features plays a crucial role in ensuring robust generalization capabilities. On the contrary, coordinate-based feature (w/o xy) somehow shows deficiency.

**Basis symbol set design.** We investigate the effect of the designs of the basis symbol set $\mathbb{S}$ on the final performance of SYMBOL. The investigation is conducted in two directions: subtracting existing symbols from $\mathbb{S}$ and augmenting them with new ones. For the former, we subtract three variables $x_*$, $\Delta x$, and $x_r$ from $\mathbb{S}$ and denote this subtracted set as $\mathbb{S}^-$. For the latter, we add three unary operators $sin$, $cos$, and $sign$ (frequently used non-linear functions in recent symbolic equation learning frameworks) to $\mathbb{S}$ and denote this augmented set as $\mathbb{S}^+$. We continue to use SYMBOL-S as a showcase and load it with $\mathbb{S}^+$ and $\mathbb{S}^-$. Under identical training and testing settings, we report the ablation results of the two ablated variants in Table 8, which show that: 1) Symbols such as $x_r$ are very important for population-based black-box optimizers, as they provide exploration behaviors for candidate populations. Removing such symbols from the symbol set significantly reduces the learning effectiveness. 2) Augmenting the symbol set certainly enlarges the representation space. However,

this addition may also affect the search for optimal update rules. This opens up a discussion for the symbolic equation learning community and ourselves on how to construct domain-specific symbol sets for extending application scenarios beyond symbolic equation learning itself.

### C.3    DISCUSSION OF PERFORMANCE OF SYMBOL-G

From experiment results in Table 1 and Figure 8, we notice that SYMBOL-G can even outperform teacher optimizers in some cases. Possible explanation are as follows. Firstly, while the reward guides the agent to closely imitate the teacher's searching behavior, our SYMBOL is not designed to merely replicate their exact update rules but to discover potentially more efficient update rules upon the basis symbol set for handling a broad spectrum of BBO task, especially with the help of our robust fitness landscape analysis features for enhanced generalization capability. Secondly, unlike traditional optimizers (e.g., MadDE), which might rely on consistent fixed searching rules, SYMBOL-G is designed to generate flexible and dynamic update rules for specific optimization steps. Lastly, we note that the inherent exploration capability of RL also plays a role in the enhanced performance. In contrast to deterministic strategies, the sampling-based strategy in RL provides more diverse actions. Such observations also show up in some imitation learning frameworks, such as the GAIL framework (Ho & Ermon, 2016). These factors collectively contribute to SYMBOL-G's ability to, in some cases, outperform its teacher optimizer. We note that by further incorporating the exploration reward (SYMBOL-E), our SYMBOL-S would consistently surpass the teacher optimizer.

