# OpenReview forum: "SYMBOL: Generating Flexible Black-Box Optimizers through Symbolic Equation Learning"
_ICLR.cc/2024/Conference — ICLR 2024 poster_

### Official Review · Reviewer_hEPw · 2023-10-30

**Soundness:** 3 good
**Presentation:** 4 excellent
**Contribution:** 3 good
**Rating:** 8
**Confidence:** 2

**Summary:**

The paper introduces SYMBOL, a novel approach for BBO that leverages a neural network trained via Reinforcement Learning to dynamically predict the explicit expression of the optimization steps for a given task.

SYMBOL is based on a Symbolic Equation Generator (SEG), an LSTM-based model responsible for predicting closed-form (symbolic) optimization updates. SEG can be trained via three different RL strategies, namely Exploration learning (SYMBOL-E), Guided learning (SYMBOL-G) and Synergized learning (SYMBOL-S). While SYMBOL-E does not impose any inductive bias about the sought-for optimizer,  SYMBOL-E explicitly forces the SEG to mimic the behaviour of a given black-box optimizer (teacher optimizer). SYMBOL-S integrates both approaches and regulates their relative importance via a hyperparameter $\lambda$.

The model -- in all its three variants -- is extensively tested and compared to multiple state-of-the-art baselines. The results indicate that SYMBOL compares favourably to the considered baselines. In addition, the updates found by SEG can be directly inspected thanks to their symbolic nature.

**Strengths:**

- The paper is well-written and the presentation is clear.

- SYMBOL represents an interesting application of deep-learning-based symbolic regression, beyond the standard setting where it is usually applied.

- The experiments compare SYMBOL to multiple baseline methods and show that SYMBOL archives state-of-the-art performance on BBO.

- Contrarily to black box systems, SYMBOL generates interpretable update steps.

**Weaknesses:**

- Reliance on a specific teacher in SYMBOL-G and SYMBOL-S: how to select a teacher to train the model may not be obvious in real-world applications. This makes the applicability of the aforementioned training strategies relatively limited.

- Reliance on the meta-learning stage: The training of the SEG model requires the selection of a training distribution D. The authors rely on 10 BBO optimization problems to construct D. However, in practice, the obtained distribution might not be large enough to guarantee sufficient generalization to fundamentally different tasks.

- Training SYMBOL is still quite computationally demanding as shown in the meta-test column in Table 1. Is training the model via RL the computational bottleneck?

**Questions:**

See weaknesses part.

---

> ### Author Response · Authors · 2023-11-18
> **Response to Reviewer hEPw**
>
> We appreciate the reviewer for the valuable feedback and the strong support for our paper. Your recognition of the paper's clarity, the novelty of SYMBOL, its state-of-the-art performance, and its good interpretability is highly encouraging. We hope that the following response, as well as the newly added results, will clear the remaining concerns.
>
> ---
>
> **[Teacher selection]** We thank the reviewer for the insightful comment. Please refer to our response **[Teacher selection in SYMBOL]** in the “general response” above.
>
> ---
>
> **[Training distribution selection]** We acknowledge the importance of task distribution $\mathbb{D}$ in facilitating generalization during the meta-learning stage of SYMBOL. To address the concern regarding the adequacy of using only 10 synthetic functions for meta-learning a universally effective policy, we have refined the description in “Training distribution $\mathbb{D}$” part in Section 4, and included additional details in Appendix B.4 of our revised paper. Our explanation is threefold:
>
> 1. The 10 synthetic functions we have selected encompass a broad spectrum of **representative** landscape features and optimisation challenges, as detailed in the newly added Table 7 in Appendix B.4. These features include uni/multi-modality, separability, smooth yet narrow ridges, and (a)symmetry, etc. In Appendix B.4, the added Figure 9 now includes visualizations of the fitness landscapes of these 10 functions. Through this diverse range and by incorporating fitness landscape analysis-based state representation (refer to Section 3.1 and Appendix A.1), SYMBOL learns not merely to solve specific tasks, but to adapt its update patterns to varying landscape features. This adaptability underpins its impressive generalization capabilities, as demonstrated in our experimental results.
> 2. We have also augmented the training distribution $\mathbb{D}$ by incorporating shifted and rotated versions of these functions, as outlined at the beginning of Section 4. This methodological augmentation enriches $\mathbb{D}$ with a multitude of diverse landscapes, thereby expanding its scope and ensuring robust generalization.
> 3. We would also like to note that these selected 10 functions have a **longstanding** history in the literature for evaluating the performance of BBO optimizers. Their proven ability to present diverse optimization properties makes them ideal for training SYMBOL, ensuring its efficacy in generalizing to a wide array of unseen, realistic tasks.
>
> ---
>
> **[Is training via RL the computational bottleneck?]** We thank the reviewer for the insightful question. We agree that a detailed examination of the computational costs of SYMBOL’s different sub-routines during training is essential. In SYMBOL, there are three main sub-routines:
>
> 1. **Update Rule Generation**: This involves computing the contextual state representation, feeding the state into the SEG (State Encoding Generator) for generating pre-order traversals, and decoding these traversals into the corresponding update rules.
> 2. **Lower-Level Optimization**: After generating the update rule, SYMBOL applies this rule to update the candidate population and evaluates the objective values of the new generation.
> 3. **Meta-Level PPO Training**: Concurrently with the lower-level optimization, the collected trajectories are utilized to update the parameters in the SEG via reinforcement learning.
>
> To quantify the computational demands, we measured the wall time for each of these sub-routines during training. The table below presents the consumed time and corresponding proportions for each subroutine, along with the total time, using SYMBOL-S as an example:
>
> | Sub-routines | Update rule generation | Lower-level optimization | Meta-level PPO training | Total time |
> | --- | --- | --- | --- | --- |
> | Time | 1.5h (14.9%) | 6.3h (62.6%) | 2.2h (22.1%) | 10h |
>
> As the data reveals, **lower-level optimization emerges as the primary computational bottleneck in SYMBOL**. It's important to clarify that high computational costs are intrinsic to BBO (Black-Box Optimization) optimization in the MetaBBO domain. This is evident in the baseline methods we compare against in our paper (like LDE, DE-DDQN, Meta-ES, etc.).
>
> To mitigate these costs, common practice involves parallel processing. Specifically, we simultaneously sample a batch of task instances and collect trajectories in parallel, significantly accelerating both the lower-level optimization and meta-level training processes. For more details on our approach to parallel processing, please refer to Appendix B.2.
>
> We hope the above clarifications help, and we are happy to engage in further discussions. Thanks for taking your time and making contributions to our paper!

---

### Official Review · Reviewer_ZLbt · 2023-10-31

**Soundness:** 3 good
**Presentation:** 4 excellent
**Contribution:** 3 good
**Rating:** 6
**Confidence:** 3

**Summary:**

This paper introduces a novel black-box optimizer (BBO) SYMBOL featured by its symbolic equation learning via meta-learning. Different from previous MetaBBOs which directly output the updated value, SYMBOL meta-learns the updated rule as an equation which achieves a superior performance and better interpretability.

**Strengths:**

1. Though the idea was also explored in various domains, I think the discoveries from this paper are still interesting. It is great to see the steps for learning more interpretable optimizers.
2. The paper is well-presented and the idea is easy to follow.
3. The Synergized learning strategy proposed in this paper is novel and interesting, which I believe could be a promising strategy in the future.
4 The empirical evaluation shows promising results against baselines. The generalization performance, thanks to the flexibility of its symbolic update rule, is quite impressive against previous MetaBBOs. The authors also carefully ablate the components of their method.

**Weaknesses:**

1. Though promising, I'm not sure if this work oversimplifies the actual optimization problem in its search space, which makes it not that useful currently. There are only 3 operators and 7 operands in its basic symbol set, is the design of the symbol set relevant to the teacher algorithm?
2. It is good to see the improvement in the experiment, however, I note SYMBOL-E does not outperform MadDe (2 out of 3 columns). This raises the question of whether the potential of SYMBOL is largely limited by the performance of MadDe. If MadDe does not work well, can the exploration reward actually help the model learn a much better update rule? The authors could try to use a relatively bad baseline as the teacher to verify this and or find a challenging task where MadDe does not work well.

**Questions:**

I'm not directly working in this field, so I'm a bit curious about how Lion [1] is compared to the symbolic discovery in this work, which I think could be more promising in the optimizer discovery.

[1] Symbolic Discovery of Optimization Algorithms. Xiangning Chen, Chen Liang, Da Huang, Esteban Real, Kaiyuan Wang, Yao Liu, Hieu Pham, Xuanyi Dong, Thang Luong, Cho-Jui Hsieh, Yifeng Lu, Quoc V. Le. https://arxiv.org/abs/2302.06675.

---

> ### Author Response · Authors · 2023-11-18
> **Response to Reviewer ZLbt (Part 1/2)**
>
> We appreciate the reviewer for the positive and valuable feedback. Thank you for acknowledging that our paper is easy to follow and that our SYMBOL is not only **flexible**, **novel**, and **interpretable**, but also showcases **promising** **performance** and **impressive** **generalization** abilities for future usage. We hope that the following response, as well as the newly added results, will clear the remaining concerns.
>
> ---
>
> **[Is the symbol set relevant to the teacher algorithm?]** We would like to clarify that the current basis symbols set is essential to cover numerous update rules (e.g., those in DEs, PSOs, GAs, etc.), and we have opted for a simple yet effective symbol set that is sufficient to generate update rules covering various searching behaviors and exhibiting different extents of exploration and exploitation abilities. Experimental results in Section 4.3, Figure 5 consistently show SYMBOL achieves new state-of-the-art performance. Meanwhile, we note that our SYMBOL can work well even if the basis symbol set is not perfectly relevant to the teacher algorithm. For example, our SYMBOL-S could outperform the teacher CMA-ES even if our symbol set could not represent the covariance update processes in CMA-ES. Please refer to our response **[Design of the symbol set]** in the “general response” above for more discussions.
>
> ---
>
> **[Whether the potential of SYMBOL is largely limited by the performance of MadDE]** Thanks for raising this concern. According to our experiments, SYMBOL-G and SYMBOL-S both outperform MadDE, clearly demonstrating the potential of SYMBOL to exceed the employed teacher optimizer. However, SYMBOL-E, designed to independently discover novel optimizers without reliance on any particular teacher optimizer, underperforms MadDE. We acknowledge that navigating the complex landscapes of BBO tasks presents significant challenges for SYMBOL-E when learning solely from scratch, which deserves further exploration in the future. So far, SYMBOL-E's strength lies in scenarios where selecting an appropriate teacher optimizer is challenging.
>
> ---
>
> **[Can the exploration reward help the model learn a much better update rule?]**  Yes, it helps. Following the suggestions by the reviewer, we address this concern from the following two perspective.
>
> **i) When the teacher optimizer is relatively bad:** In Section 4.3, we explored the use of a relatively less effective teacher, vanilla DE, within SYMBOL-S. Despite the limitations of the teacher optimizer, SYMBOL-S consistently outperformed it, as evidenced by the performance curves in Figure 5 (Section 4.3) and Figure 8 (Appendix C.1). This indicates the robustness of SYMBOL-S in surpassing teacher capabilities.
>
> **ii) When the optimization task is challenging for the teacher optimizer**: In the context of more challenging tasks such as HPO-B and Protein-docking (Table 1, Meta Generalization part), SYMBOL exhibits impressive performance even when the teacher optimizer (MadDE) struggles. Both SYMBOL-E and SYMBOL-G show only limited improvements when used independently, yet SYMBOL-S significantly outperforms MadDE, underscoring that SYMBOL is not constrained by its teacher's performance. The synergy between exploration and imitation rewards in SYMBOL not only demonstrates robust generalization across diverse tasks but also ensures rapid convergence.

---

> > ### Author Response · Authors · 2023-11-18
> > **Response to Reviewer ZLbt (Part 2/2)**
> >
> > **[Difference between SYMBOL and Lion]**  We thank the reviewer for pointing out the work of Lion. While we acknowledge Lion's success in auto-discovering gradient descent optimizers, it is not suitable for BBO optimization due to the following differences between SYMBOL and Lion:
> >
> > 1. **Targeted Domain**: Lion is designed to generate gradient descent optimizers, whereas SYMBOL focuses on discovering novel update rules for black-box optimization tasks.
> > 2. **Search Space**: Lion operates within a "Program search space" comprising 43 functions derived from Numpy/JAX and first-order optimization algorithms, many of which are not applicable to the black-box optimization context as in our work (e.g., global_norm and dot product functions). SYMBOL, in contrast, searches effective mathematical expressions (update rules) in the symbolic expression space expanded from the constructed symbol set with a more compact set of 10 operators and operations tailored for discovering novel BBO update rules.
> > 3. **Training Strategy**: Lion utilizes regularized evolution involving a large population of algorithms improved via mutation-selection cycles, demanding substantial computational resources (100 TPU v2 chips for 72 hours). SYMBOL adopts a more resource-efficient approach using policy gradient methods (PPO), enabling the meta-learning of high-reward update rules with just 1 RTX-3090 GPU in 10 hours.
> > 4. **Discovered Algorithm**: Lion outputs a single robust gradient descent algorithm, less suited for BBO contexts where dynamic adaptation of update rules is crucial. SYMBOL, conversely, discovers a SEG model capable of generating appropriate update rules dynamically throughout the optimization process.
> >
> > In the revised paper (Section 2), we have mentioned Lion as a related work. Regarding a direct comparison between SYMBOL and Lion, we would like to note that:
> >
> > - The full details of Lion's training framework are not publicly available (see AutoML official GitHub repositories), limiting our ability to implement and conduct a fair comparison.
> > - The resource requirements for training Lion (100 TPU v2 chips, 72 hours) contrast sharply with SYMBOL's more modest needs (1 RTX-3090 GPU, 10 hours), making a timely implementation challenging.
> >
> > We hope the above clarifications help, and we are happy to engage in further discussions. Thanks for taking your time and making contributions to our paper!

---

> > > ### Comment · Reviewer_ZLbt · 2023-11-22
> > > **Thanks for the response**
> > >
> > > I want to thank the authors for addressing my concerns. I have carefully read the response, I think my main concern is about the relationship between SYMBOL's performance and MadDE's performance. I accept the author's response and I will keep my score.

---

> > > > ### Author Response · Authors · 2023-11-22
> > > > **Thanks for Reviewer ZLbt**
> > > >
> > > > We are glad to have your concern addressed. Once again, we want to convey our sincere appreciation for your suggestions to enhance our paper.

---

### Official Review · Reviewer_VT1T · 2023-11-01

**Soundness:** 3 good
**Presentation:** 3 good
**Contribution:** 3 good
**Rating:** 6
**Confidence:** 1

**Summary:**

This paper proposes SIMBOL, a framework to learn a black-box optimizer through symbolic equation learning. The paper first presents a symbolic equation generator (SEG) to generates closed-form optimization rule, where such closed-form is found through reinforcement learning. The paper argues the proposed method shows state-of-the-art results on benchmarks as well as showing  strong zero-shot generalization capabilities.

**Strengths:**

- Compared with existing BBO, the proposed method seems technically novel (I am not the expert).
- The paper is generally well-written and easy-to-follow.
- The results seem promising compared with existing methods.

**Weaknesses:**

- There's no strategy in choosing a teacher optimizer ($\kappa$).
- Since it requires training based on reinforcement learning, training may require a time for framework compared with existing BBO that does not require any training. I wonder how long it takes for training compared with other MetaBBO method?

**Questions:**

I have no expertise in this area, so I will adjust my score based on discussion between authors reviewers, and AC.

---

> ### Author Response · Authors · 2023-11-18
> **Response to Reviewer VT1T**
>
> We appreciate the reviewer for the positive and valuable feedback. Thank you for acknowledging that our paper is well-written and that our proposed SYMBOL is novel and promising. We hope that the following response will clear the remaining concerns.
>
> ---
>
> **[Strategy in choosing a teacher optimizer]** We thank the reviewer for the insightful comment. Please refer to our response **[Teacher selection in SYMBOL]** in the “general response” above.
>
> ---
>
> **[Training cost comparison between SYMBOL and baselines]** We would like to clarify that the training time of SYMBOL and other MetaBBO baselines have been reported in Section 4.1, Table 1. For your convenience, we list the training time in the table below.
>
> | SYMBOL-E | SYMBOL-G | SYMBOL-S | LDE | DEDDQN | Meta-ES | MELBA | RNN-Opt |
> | --- | --- | --- | --- | --- | --- | --- | --- |
> | 6h | 10h | 10h | 9h | 38m | 12h | 4h | 11h |
>
> We hope the above clarifications help, and we are happy to engage in further discussions. Thanks for taking your time and making contributions to our paper!

---

> > ### Comment · Reviewer_VT1T · 2023-11-22
> > **Response**
> >
> > Thanks for the response. I've read the other reviews' as well and I think this is a valuable work (though I am not en expert in this area). Thus, I retain my score.

---

> > > ### Author Response · Authors · 2023-11-23
> > > **Thanks for Reviewer VT1T**
> > >
> > > Thank you for agreeing with our work. Once again, we sincerely appreciate the valuable time and effort you have invested in reviewing our paper.

---

### Official Review · Reviewer_h4Ki · 2023-11-08

**Soundness:** 3 good
**Presentation:** 3 good
**Contribution:** 2 fair
**Rating:** 6
**Confidence:** 3

**Summary:**

This paper proposes meta-learning for black-box optimization (BBO) methods using symbolic equation learning. The proposed method, termed SYMBOL, trains the neural network model that generates the update rule of the BBO method depending on the task and optimization situation using landscape features as the input of the model. The model for generating the update rule is trained based on the reinforcement learning algorithm. The experimental evaluation using artificial benchmark functions, HPO, and Protein docking benchmarks demonstrates that the optimizer generated by the proposed SYMBOL can beat the existing BBO and MetaBBO baselines.

**Strengths:**

- The proposed SYMBOL can dynamically change the update rule of solutions depending on the search situation owing to the use of fitness landscape features, which seems to be a technical novelty. In addition, the training strategy of the optimizer generator by mimicking the existing teacher black-box optimizer is reasonable for accelerating the model training.
- The search performance of the optimizer generated by SYMBOL outperforms other BBO and recent MetaBBO techniques.

**Weaknesses:**

- It is somewhat unclear to me the key difference and novelty of the proposed SYMBOL because there exists a lot of MetaBBO techniques. It would be very nice if the authors clarified the advantages and key differences of the SYMBOL compared to other existing methods.
- Although the proposed SYMBOL can generate flexible update rules of solutions, the representation ability of update rules is limited depending on the pre-defined basic symbol set. I suppose that SYMBOL cannot generate the CMA-ES type update rules.

**Questions:**

- Many techniques regarding MetaBBO, including meta genetic algorithm and meta evolutionary algorithm, have been developed so far [i]. It might be better to mention the traditional approaches related to MetaBBO.
- The work of [ii] might relate to this paper. Although it only tunes the step-size adaptation in CMA-ES, the concept and used techniques, such as reinforcement learning-based training and guided policy search, are somewhat similar.
- Why does SYMBOL-G outperform MadDE in Table 1? As the teacher optimizer of SYMBOL-G is MadDE in the experiment, it seems strange that SYMBOL-G beats the teacher optimizer.
- Could you describe the relationship between MetaBBO and automatic algorithm configurations?

[i] Qi Zhao, Qiqi Duan, Bai Yan, Shi Cheng, Yuhui Shi, "A Survey on Automated Design of Metaheuristic Algorithms," arXiv:2303.06532

[ii] Shala, G., Biedenkapp, A., Awad, N., Adriaensen, S., Lindauer, M., Hutter, F. (2020). Learning Step-Size Adaptation in CMA-ES. In: Bäck, T., et al. Parallel Problem Solving from Nature – PPSN XVI. PPSN 2020. Lecture Notes in Computer Science(), vol 12269. Springer, Cham. https://doi.org/10.1007/978-3-030-58112-1_48

Minor comments:
- On page 2, "At the lower lever" should be "At the lower level."
- In equation (3), the redundant right-side parenthesis exists.


----- After the rebuttal -----

Thank you for the responses. As the responses are convincing, I keep my score on the acceptance side.

---

> ### Author Response · Authors · 2023-11-18
> **Response to Reviewer h4Ki (Part 1/2)**
>
> We appreciate the reviewer for the positive and valuable feedback. Thank you for acknowledging that our SYMBOL is technically novel, reasonable, and could outperform other BBO and MetaBBO methods. We hope that the following response, as well as the newly added results, will clear the remaining concerns.
>
> ---
>
> **[Clarify the advantages and key differences of our SYMBOL]** We thank the reviewer for the suggestion and apologize for any confusion. **To the best of our knowledge,** **SYMBOL represents the initial endeavour to promote the automated discovery of BBO update rules through symbolic equation learning**. Compared to existing MetaBBO techniques, our SYMBOL is **more flexible** (can generate novel update rules beyond hand-crafted rules dynamically depending on the search situation), **more generalizable** (owing to the leveraged fitness landscape analysis as robust state features), and **more interpretable** (the generated update rules are in closed form). Meanwhile, our SYMBOL archives new state-of-the-art performance among other BBO and recently developed MetaBBO methods. Concerning other MetaBBO works, two primary branches exist — one for auto-configuration and the other for candidate solution proposals, which are discussed as follows:
>
> 1. **MetaBBO for Auto-Configuration:** This branch involves a neural policy at the meta level that dictates configurations for the **hand-crafted update rules** in the lower-level optimizer, as seen in Table 1 with baselines like LDE, MELBA, and Meta-ES. Alternatively, some other works leverage the meta-level agent to select a proper **hand-crafted update rule** from a predetermined pool of rules, as done by the DEDDQN in Table 1. However, these methods may inherit potential limitations from the hand-crafted rules within the optimizer itself. SYMBOL minimizes dependence on hand-crafted rules and flexibly generates closed-form update rules upon a symbol set, enhancing the exploration capability of efficient update rules and consequently boosting the potential performance of the generated optimizers.
> 2. **MetaBBO for Candidate Solution Proposal:** This branch employs RNNs to auto-regressively propose the next generation of candidates end-to-end without explicit update rules. Specifically, the input is the position of the current candidate(s), and the output is the next candidate(s). However, these models may easily get overfitted, and their efficiency may be limited to low-dimensional BBOs only, raising generalization concerns. In contrast, SYMBOL incorporates fitness landscape analysis into the state representation, aiming to construct a generalizable model that can produce flexible update rules for various problems, ensuring a more robust optimization. Additionally, while existing works for candidate proposals are entirely "black-box" with limited interpretability, SYMBOL's generated rules are step-by-step transparent, allowing for easier analysis and understanding of the learned processes.
>
> Correspondingly, we have refined our discussions in the second paragraph of the **Introduction** section and the second paragraph of the **Related Works** section.
>
> ---
>
> **[Representation ability of update rules]** We thank the reviewer for the insightful comment. Please refer to our response **[Design of the symbol set]** in the “general response” above.
>
> ---
>
> **[Q1: Mention traditional MetaBBO works]** We thank the reviewer for the suggestion. We have enriched the **Related Work** section. As introduced in [i], these traditional methods also adhere to the bi-level optimization paradigm. But in their paradigm, both the meta-level and lower-level components are implemented using traditional black-box optimizers. In addition, they generate a single strategy for the entire lower-level optimization process. Our SYMBOL differs in that a) it employs a learning-based RL-Agent as the meta-level method; b) it generates flexible update rules in the lower level; and c) it dynamically adjusts its strategy throughout the lower-level optimization process, adapting to changing conditions and ensuring greater flexibility.

---

> > ### Author Response · Authors · 2023-11-18
> > **Response to Reviewer h4Ki (Part 2/2)**
> >
> > **[Q2: Add LTO-CMA]** We thank the reviewer for pointing out this reference. We acknowledge that LTO-CMA [ii] is related to SYMBOL in concept, as both approaches aim to search for the optimal policy by leveraging reinforcement learning. However, as the reviewer has pointed out, LTO-CMA is confined to adapting the step-size for CMA-ES, while our SYMBOL generates the update rule from scratch. Additionally, the state representation of LTO-CMA includes CMA-ES-specific entries, making it less generic. In contrast, SYMBOL provides more flexibility in terms of teacher options, as demonstrated in the experimental section of the paper (Section 4.3, Figure 5). Moreover, there are other distinctions between LTO-CMA and our SYMBOL, including the usage of teachers, the learning objective, and the learning algorithm.
> >
> > In our meta-test and meta-generalization experiments, we further evaluated LTO-CMA on the $\mathbb{D}$, HPO-B and Protein-docking set by directly loading the trained model in their Github Repo. We then compared its performance with our SYMBOL-S. For a fair comparison, we used SYMBOL-S with CMA-ES as the teacher, trained in Section 4.3. The table below reports the average of min-max normalized performance with standard deviation over 5 independent runs. The results underscore that MetaBBO for auto-configuration may inherit limitations from its backbone optimizer, while our SYMBOL generates flexible and novel update rules that generalize well to unseen tasks.
> >
> > |  | CMA-ES | LTO-CMA | SYMBOL-S |
> > | --- | --- | --- | --- |
> > | D | 0.944 (0.018) | 0.956 (0.016) | 0.966 (0.008) |
> > | HPO-B | 0.917 (0.003) | 0.920 (0.005) | 0.929 (0.020) |
> > | Protein-docking | 0.979 (0.000) | 0.984 (0.000) | 0.997 (0.000) |
> >
> > We have added LTO-CMA as a related work in Section 2.
> >
> > ---
> >
> > **[Q3: Why does SYMBOL-G outperform MadDE?]** We understand your concern regarding the relatively superior performance of SYMBOL-G compared to MadDE, considering MadDE as the teacher optimizer. Firstly, while the reward guides the agent to closely imitate the teacher's searching behavior, our SYMBOL is not designed to merely replicate their exact update rules but to discover potentially more efficient update rules upon the basis symbol set for handling a broad spectrum of BBO task, especially with the help of our robust fitness landscape analysis features for enhanced generalization capability. Secondly, unlike traditional optimizers (e.g., MadDE), which might rely on consistent fixed searching rules, SYMBOL-G is designed to generate flexible and dynamic update rules for specific optimization steps. Lastly, we note that the inherent exploration capability of RL also plays a role in the enhanced performance. In contrast to deterministic strategies, the sampling-based strategy in RL provides more diverse actions. Such observations also show up in some imitation learning frameworks, such as the GAIL framework [1]. These factors collectively contribute to SYMBOL-G's ability to, in some cases, outperform its teacher optimizer. We note that by further incorporating the exploration reward (SYMBOL-E), our SYMBOL-S would consistently surpass the teacher optimizer. We have included the above discussion in Appendix C.3 in the revised PDF.
> > ```
> > References:
> >
> > [1] Jonathan Ho, Stefano Ermon, "Generative Adversarial Imitation Learning", arXiv: 1606.03476
> > ```
> >
> > ---
> >
> > **[Q4: Relationship between MetaBBO and AutoAC]** MetaBBO and automatic algorithm configurations (AutoAC) define new or improved optimization approaches from different perspectives: MetaBBO aims to meta-learn BBO optimizer that achieves boosted performance on BBO tasks, while AutoAC aims to fine-tune the configuration of the backbone algorithm. However, there exists an intersection of the two domains: when one method follows the bi-level optimization paradigm with the meta-level optimizer automatically fine-tuning the configuration of a lower-level optimizer to boost its performance on BBO tasks, then this method can belong to both MetaBBO and AutoAC. However, when limiting the scope to the BBO field, the scope of MetaBBO is larger than that of AutoAC in general. Besides, recent MetaBBO includes more approaches, such as directly generating the update rules like SYMBOL.
> >
> > ---
> >
> > **[Minor comments on typos]** We apologize for any confusion caused by typos in our previous submission. We have made the necessary corrections and have thoroughly proofread the revised manuscript.
> >
> > We hope the above clarifications help, and we are happy to engage in further discussions. Thanks for taking your time and making contributions to our paper!

---

### Author Response · Authors · 2023-11-18
**General Response (Part 1/2)**

We thank all the reviewers for their valuable comments. We are pleased to see that **all the reviewers recommend the acceptance of our paper**, and the reviewers have recognized our SYMBOL to be technically novel (#**h4Ki, #VT1T, #ZLbt**), promising (**#VT1T, #ZLbt**),  well-written (**#VT1T, #ZLbt, #hEPw**), flexible (**#ZLbt**), generalizable (**#ZLbt**), and interpretable (**#ZLbt, #hEPw**) MetaBBO framework achieving state-of-the-art performance (**all reviewers**). In this general response, we intend to address two common concerns. Note that all revisions for all reviewers are colored in red in the revised PDF.

---

---

> ### Author Response · Authors · 2023-11-18
> **General Response (Part 2/2)**
>
> **[Design of the symbol set]**  Reviewers #**h4Ki** and **#ZLbt** raised concerns regarding the representation ability of our symbol set. We value this insightful feedback and agree with the reviewers that the current basis symbol set could cover most essential update rules (e.g., those in DEs, PSOs, GAs, etc.) but not all complex ones (e.g., the covariance update processes in CMA-ES). However, we would like to clarify the following points:
>
> 1. The primary motivation of SYMBOL is to dynamically generate BBO update rules from scratch in a data-driven fashion, reducing dependence on conventional, intricate update rules. With the incorporation of teacher guidance for accelerated convergence in our SYMBOL-G and SYMBOL-S, our aim is not to directly replicate the specific complex update rules of teacher optimizers. Instead, we seek to emulate their search behaviors. This allows SYMBOL to discover unique, simpler, yet highly effective optimizers that can compete with, but not necessarily replicate, sophisticated optimizers like CMA-ES. As we stated in Section 3.1, our symbol set could cover numerous update rules (e.g., those in DEs, PSOs, GAs, etc.), but more importantly, could span beyond the existing hand-crafted space, offering flexibility to explore novel and effective update rules. Our meta-test and meta-generalization results demonstrate that effective update rules can be meta-learned from this symbol set, and new state-of-the-art optimization performance is achieved against the baselines.
> 2. Despite the current symbol set's inability to generate CMA-ES-type update rules, our study demonstrates that this simple symbol set is sufficient to generate update rules covering various searching behaviors and exhibiting different extents of exploration and exploitation abilities. Experimental results in Section 4.3, Figure 5 consistently show SYMBOL outperforming many teacher optimizers, including CMA-ES.
> 3. We acknowledge the challenge of designing an ideal symbol set. There is a balance between having enough symbols to represent complex equations and maintaining good equation construction efficiency. Our ablation studies, detailed in Table 9 and Appendix C.2, delve into the impact of altering the symbol set on SYMBOL's learning efficacy. As the first research effort on generative MetaBBO through symbolic equations, we have opted for a simple yet effective symbol set, and leave the exploration of more powerful symbol sets as important future works. Correspondingly, we have carefully refined our claims regarding the symbol set in **Section 3.1** and outlined potential future works regarding this in the **Conclusion** section.
>
> ---
>
> **[Teacher selection in SYMBOL]** Reviewers #**VT1T** and #**hEPw** raised questions regarding the teacher selection in our SYMBOL-G and SYMBOL-S. We would like to offer some clarifications:
>
> 1. **SYMBOL is teacher-agnostic**. Our model architecture, as detailed in Section 3.1, uses a contextual state representation that requires no information about the teacher optimizer. This design makes SYMBOL a generic, plug-and-play framework compatible with various teacher optimizers. Our experiments (Section 4.3 and Appendix C.1) demonstrate SYMBOL's effectiveness across different teacher optimizers (including DE, PSO, CMA-ES), where the performance consistently outperforms each teacher.
> 2. **In most cases, selecting a teacher for SYMBOL is not hard**.
> - As we did in SYMBOL, one may select a proper optimizer that has been reported to have excellent performance in existing BBO benchmarks (e.g., COCO, IEEE CEC Competition Series). Note that such BBO benchmarks are proposed to examine an optimizer’s potential for complex, realistic tasks. In this paper, MadDE is chosen as the teacher optimizer since it shows superior optimization performance in our preliminary numerical tests.
> - One may also seek proper teacher optimizers from the state-of-the-art results reported in the latest literature on targeted application scenarios. However, this requires expert-level knowledge and continuous follow-up in that domain.
> 3. **Fallback Strategy: SYMBOL-E.** Acknowledging the potential challenges in teacher selection for emerging optimization tasks, we introduced SYMBOL-E. This strategy allows for meta-learning an effective policy from scratch, without relying on any teacher optimizer, thus providing a feasible alternative when the teacher optimizer is strictly not available.

---

### Public Comment · ~Kai_Wu3 · 2023-11-21

Good work in interpretable MetaBBO through symbolic equation learning. I think the following comments could be used to enhance this manuscript.

1. This work lacks a discussion regarding the distinctions between SYMBOL and related efforts like

[1] "Symbolic Learning to Optimize: Towards Interpretability and Scalability" (https://arxiv.org/abs/2203.06578).

2. Notably, this work overlooks significant efforts in meta-learned black-box optimizers, such as:

[2] "Learning to Optimize in Swarms," NeurIPS 2019

[3] "B2Opt: Learning to Optimize Black-box Optimization with Little Budget" (https://arxiv.org/abs/2304.11787)

[4] "DECN: Automated Evolutionary Algorithms via Evolution Inspired Deep Convolution Network" (https://arxiv.org/pdf/2304.09599.pdf)

3. There's disagreement with the claim that many MetaBBO works fail to generalize across different task distributions, dimensions, population sizes, or optimization horizons. [3] and [4] demonstrate capabilities in handling these cases.

4. The problem's dimension in this study is relatively small, approximately 10 or 12. Thus, what advantages does this method offer over others concerning problem dimension?

---

> ### Author Response · Authors · 2023-11-21
> **Response to Public Comments from Kai Wu**
>
> Thank you for your interest in and recognition of SYMBOL, as well as for participating in the discussion. We highly value your comments and would like to provide the following clarification.
>
> 1. The distinctions between [1] and SYMBOL are summarized as follows:
>
> - Although both utilize symbolic regression techniques, the frameworks of [1] and SYMBOL are markedly different. Specifically, [1] follows a two-phase distillation framework: it first trains a numerical L2O model (such as RNN-Opt in our paper's baselines) by maximizing its performance on the training tasks, and then utilizes a GP-based symbolic regression method to search for symbolic update rules that best match the input and output pairs generated by the meta-learned L2O model for distillation. In contrast, SYMBOL meta-learns novel and effective update rules in a unified, end-to-end manner without a second phase of searching. Notably, SYMBOL is positioned as a generative approach rather than a distillation framework.
> - The targeted tasks of [1] and SYMBOL are different. Both phases in [1] rely on gradient information and are dedicated to discovering a gradient descent algorithm. On one hand, it's essential to note that gradient information is not accessible for the black-box optimization (BBO) tasks we focus on, whereas SYMBOL incorporates gradient-free state representation as the input to generate BBO update rules. On the other hand, SYMBOL auto-regressively generates flexible update rules at each optimization step according to different optimization statuses, without limiting its behavior like a gradient descent algorithm.
>
>
> ---
> 2. The studies [2]-[4] have a similar methodology to the RNN-Opt [5] compared in our paper. These works require gradient information during training and limit the training problem sets to be differentiable. However, our SYMBOL approach focuses on more general scenarios where gradients cannot be used during training. In our paper, we compare our SYMBOL to more related works, such as DEDDQN[6], LDE[7], and Meta-ES[8], in the experimental varification. We also include RNN-Opt [5] as an external reference, and our results show that it generally underperforms our SYMBOL as well as [6]-[8]. Nevertheless, we will include these valuable works into our final version of paper, in the Related Works section.
> ---
> 3. Among the four works you mentioned, [2] has difficulty generalizing across different problem dimensions; [3] cannot generalize across different population sizes due to the pre-defined entries of parameters; and both [3] and [4] require re-training on unseen tasks, limiting their generalization across different tasks. In comparison, our SYMBOL approach is carefully designed to overcome these limitations and generalize effectively across various task distributions, dimensions, population sizes, and optimization horizons, as illustrated in Table 1 in Section 4.1 of our paper.
>
> ---
> 4. Regarding the problem dimensions, we would like to clarify:
> - Although the COCO benchmarks, HPO tasks and Protein-Docking tasks in our paper possess no more than $20$ dimensions, their intricate optimization properties challenge the searching ability of BBO optimizers. They are widely used for evaluating BBO optimizers and MetaBBO methods (such as the baselines adopted in our paper: Meta-ES[8], MelBa[9], etc). The results in our paper show that SYMBOL achieves significant improvement against the MetaBBO baselines.
> - From our aspects, a potential advantage of SYMBOL concerning different problem dimensions locates at the utilization of fitness landscape analysis based features as input, which is a fixed dimensional vector. Compared against the coordinate-wised input in [1]-[4], this design makes the SEG network in SYMBOL easily generalized to solve high dimensional tasks without additional overhead for processing the input.
>
>
>
> ---
> We hope the above responses addressed your concerns.
>
> [1] "Symbolic Learning to Optimize: Towards Interpretability and Scalability" (https://arxiv.org/abs/2203.06578).
>
> [2] "Learning to Optimize in Swarms" (https://arxiv.org/abs/1911.03787).
>
> [3] "B2Opt: Learning to Optimize Black-box Optimization with Little Budget" (https://arxiv.org/abs/2304.11787)
>
> [4] "DECN: Automated Evolutionary Algorithms via Evolution Inspired Deep Convolution Network" (https://arxiv.org/pdf/2304.09599.pdf)
>
> [5] “Meta-Learning for Black-box Optimization”, (https://arxiv.org/abs/1907.06901).
>
> [6] “Deep reinforcement learning based parameter control in differential evolution”, (https://dl.acm.org/doi/abs/10.1145/3321707.3321813).
>
> [7] “Learning adaptive differential evolution algorithm from optimization experiences by policy gradient”, (https://arxiv.org/abs/2102.03572).
>
> [8] “Discovering Evolution Strategies via Meta-Black-Box Optimization”, (https://dl.acm.org/doi/abs/10.1145/3583133.3595822).
>
> [9] “Meta-learning of Black-box Solvers Using Deep Reinforcement Learning”, (https://openreview.net/forum?id=9pO8hSVu0J).

---

> > ### Public Comment · ~Kai_Wu3 · 2023-12-05
> >
> > Thanks. All my comments have been addressed. Good work!

---

### Meta-Review · Area_Chair_jmTU · 2023-12-05

**Metareview:**

This paper proposes a new method, SYMBOL (and variants) in the field of learned blackbox optimization and genetic algorithms. In summary, the method seeks to update an entire population by generating an update rule. Specifically, the architecture:
* Takes in as input the contextual state of the population, consisting of essentially normalized features such as average individual distances, objective gap against best-so-far, and time-stamp of generation.
* Performs a sequential loop (ex with LSTM) to generate tokens, representing generational update rules.

Training was performed policy gradients, with the reward based on a mix of objective performance and behavioral cloning. Pretraining was performed over BBOB data, and evaluations were done over BBOB, HPOB, and Protein Docking tasks. Visualizations over exact symbols and population updates were given when optimizing over a 2D Rastrigin function.

Overall, the paper's ideas are reasonable and experimental results are solid (as mentioned by multiple reviewers).

One common complaint raised by nearly all reviewers is the limited symbol set, which only consists of basic addition/subtraction/etc and numerical tokens. Thus all outputs of SYMBOL will look like $x + 0.18 \times (x^{\*} - x_{r}) + 0.42 \times (x_{i}^{\*} - x_{r})$, and it is questionable as to how truly interpretable these types of outputs are, and whether these resulting algorithms will lead to broad usage.

This should be investigated in a further paper, but for now my recommendation is to accept.

**Justification For Why Not Higher Score:**

The question to ask is whether the work leads to more broad insights or impactful usage (e.g. do the symbols generated teach us new techniques of designing algorithms?). Take for example, the Lion paper (https://arxiv.org/abs/2302.06675) in which a discovered optimizer was indeed broadly used among the community.

Currently, the case is unclear for this SYMBOL paper - I don't believe we really learn any new profound ways of designing algorithms from looking at expressions of the form $x + 0.18 \times (x^{\*} - x_{r}) + 0.42 \times (x_{i}^{\*} - x_{r})$.

Thus I can for now only recommend poster accept.

**Justification For Why Not Lower Score:**

The paper is well-written, the proposed method makes sense, and the experimental results are solid (an opinion shard by all reviews). This at least deserves a poster accept.

---

### Decision · Program_Chairs · 2024-01-16

Accept (poster)